# Repression of PRMT activities sensitize human homologous recombination-proficient ovarian and breast cancer cells to PARP inhibitor treatment

Youyou Zhang[1,2†‡], Mu Xu[1†], Jiao Yuan[1], Zhongyi Hu[1], Junjie Jiang[1], Yanrong Sun[1], Jie Huang[1], Yuxin Wang[1], Bingwei Wang[1], Jianfeng Shen[1], Meixiao Long[3], Yi Fan[4,5], Kathleen T Montone[6], Janos Tanyi[2,4,7], Sarah H Kim[2,4], Omid Tavana[8], Robert H Vonderheide[4], Ho Man Chan[8], Susan Domchek[4,9,10], Lin Zhang[1,2,4*], Xiaowen Hu[1,2*]

[1]Center for Women's Health and Reproductive Medicine, University of Pennsylvania, Philadelphia, United States; [2]Department of Obstetrics and Gynecology, University of Pennsylvania, Philadelphia, United States; [3]Division of Hematology, Department of Internal Medicine, Ohio State University, Columbus, United States; [4]Abramson Cancer Center, University of Pennsylvania, Philadelphia, United States; [5]Department of Radiation Oncology, University of Pennsylvania, Philadelphia, United States; [6]Department of Pathology and Laboratory Medicine, University of Pennsylvania, Philadelphia, United States; [7]Center for Gynecologic Cancer Immunotherapies, University of Pennsylvania, Philadelphia, United States; [8]Bioscience, Research and Early Development, Oncology R&D, AstraZeneca, Waltham, United States; [9]Division of Hematology and Oncology, Department of Medicine, University of Pennsylvania, Philadelphia, United States; [10]Basser Center for BRCA, University of Pennsylvania, Philadelphia, United States

**\*For correspondence:**
linzhang@pennmedicine.upenn.edu (LZ);
xiaowenh@pennmedicine.upenn.edu (XH)

[†]These authors contributed equally to this work

**Present address:**
[‡]GlaxoSmithKline, Collegeville, United States

## eLife Assessment

This study presents a **valuable** and interesting finding that a combination of arginine methyltransferase inhibitors synergize with PARP inhibitors to eliminate ovarian and triple negative breast cancer cell lines in vitro and in vivo using preclinical mouse models. The data were collected and analyzed using **solid** and validated methodology and can be used as a starting point for the development of novel therapeutics. The work will be of broad interest to scientists working in the field of breast cancer and ovarian cancer.

**Abstract** Therapeutic epigenetic modulation is currently being evaluated in the clinic to sensitize homologous recombination (HR)-proficient tumors to PARP inhibitors. To broaden its clinical applicability and identify more effective combination strategies, we conducted a drug screen combining PARP inhibitors with 74 well-characterized epigenetic modulators targeting five major classes of epigenetic enzymes. Notably, both type I PRMT inhibitors and PRMT5 inhibitors scored highly in combination efficacy and clinical prioritization. PRMT inhibition significantly enhanced PARP inhibitor-induced DNA damage in human HR-proficient ovarian and breast cancer cells. Mechanistically, PRMT suppression downregulates DNA damage repair genes and BRCAness-associated pathways, while also modulating intrinsic innate immune responses within cancer cells. Integrative

analysis of large-scale genomic and functional datasets from TCGA and DepMap further supports PRMT1, PRMT4, and PRMT5 as promising therapeutic targets in oncology. Importantly, dual inhibition of PRMT1 and PRMT5 synergistically sensitizes tumors to PARP inhibitors. Collectively, our findings provide strong rationale for the clinical development of PRMT and PARP inhibitor combinations in HR-proficient ovarian and breast cancers.

## Introduction

High-grade serous ovarian cancer (HGSOC) and triple-negative breast cancer (TNBC) share clinical and genomic characteristics, such as poor prognosis, homologous recombination (HR) deficiencies, and potential immunoreactivity (*Cancer Genome Atlas Research Network, 2011*; *Cancer Genome Atlas, 2012*; *Bowtell et al., 2015*; *Bianchini et al., 2022*). There is an urgent unmet medical need to develop effective therapeutic strategies for these two diseases. Recently, limited targeted-therapy options, such as the Poly (ADP-ribose) polymerase inhibitors (PARPis), have been approved to treat HGSOC and TNBC in a frontline setting as maintenance therapies or in a recurrent/metastatic setting (*Scott et al., 2015*; *Pommier et al., 2016*; *Lord and Ashworth, 2017*; *Dias et al., 2021*; *Konstantinopoulos et al., 2015*; *Pilié et al., 2019*). However, the greatest clinical benefit from PARPi monotherapy has been mainly observed in HR-deficient tumors (*Scott et al., 2015*; *Pommier et al., 2016*; *Lord and Ashworth, 2017*; *Dias et al., 2021*; *Konstantinopoulos et al., 2015*; *Pilié et al., 2019*), only representing a minority of patients (less than 50% of patients even in HGSOC *Cancer Genome Atlas Research Network, 2011*; *Bowtell et al., 2015*). Due to frequent defection of one or more DNA damage repair (DDR) pathways, increased replication stress, and higher endogenous DNA damage levels, cancer cells highly rely on the DDR pathways for survival compared with normal cells (*Pilié et al., 2019*; *Curtin, 2012*; *O'Connor, 2015*; *Pearl et al., 2015*; *Groelly et al., 2023*). During tumorigenesis, to solve these crises, certain DDR-related genes are pathologically upregulated in cancer cells. Epigenetic mechanisms represent a major strategy used by cancer cells to maintain or restore their abnormal need for DDR gene expression, resulting in a targetable vulnerability (*Figure 1A*). This 'Achilles' heel' can be targeted by epigenetic modulators (e.g. BETi *Yang et al., 2017*; *Karakashev et al., 2017*; *Sun et al., 2018*, CDK7/12i *Johnson et al., 2016*; *Iniguez et al., 2018*; *Shan et al., 2020*, DNMTi *Muvarak et al., 2016*; *Abbotts et al., 2019*; *Baer et al., 2022*, and HDACi *Adimoolam et al., 2007*; *Konstantinopoulos et al., 2014*) or small molecules that indirectly modulate DDR gene transcription (e.g., PI3KCAi *Ibrahim et al., 2012*; *Juvekar et al., 2012*; *Konstantinopoulos et al., 2019*; *Batalini et al., 2022*, AKTi *Yap et al., 2020*; *Westin et al., 2021*, mTORi *Mo et al., 2016*, and MEKi *Sun et al., 2017*, *Figure 1B*). Several groups, including our own (*Yang et al., 2017*), have developed an 'induced PARPi sensitivity by epigenetic modulation' strategy (*Konstantinopoulos et al., 2015*; *Pilié et al., 2019*; *Yang et al., 2017*) to preferentially impair the transcription of certain DDR genes, thereby sensitizing PARPi intrinsic and acquired resistant tumors to PARPi treatment. For example, our previous discovery on the combination of BETi and PARPi (*Yang et al., 2017*) is being evaluated in a Phase 1b/2 trial and achieving promising clinical results (NCT03901469; *Aftimos et al., 2022*). Excitingly, due to cancer cells being extremely sensitive to the epigenetic repression of DDR genes compared with normal cells, a therapeutic window for these combinations has been observed during early clinical development (*Baer et al., 2022*; *Konstantinopoulos et al., 2019*; *Batalini et al., 2022*; *Yap et al., 2020*; *Westin et al., 2021*; *Aftimos et al., 2022*). Although promising results have been reported in the clinic, the following two key questions still need to be addressed: First, among the clinically applicable epigenetic modulators, which epi-drug(s) can provide the strongest synergistic effects with PARPi? Second, can a BRCA-independent strategy be developed, given that most current strategies are mediated by repressing the expression of BRCA, which has been genomically mutated in acquired HR-proficient HGSOC/TNBC patients. Therefore, a highly effective and BRCA-independent strategy needs to be designed and evaluated.

Protein arginine methyltransferases (PRMTs) have emerged as attractive therapeutic targets in oncology owing to their crucial functions during tumorigenesis (*Blanc and Richard, 2017*; *Guccione and Richard, 2019*; *Jarrold and Davies, 2019*; *Lorton and Shechter, 2019*; *Wu et al., 2021*). Arginine methylation is a common post-translational modification of proteins, playing crucial functions in the regulation of transcription (*Migliori et al., 2012*; *Cheung et al., 2016*; *Chiang et al., 2017*; *Pastore et al., 2020*), splicing (*Bezzi et al., 2013*; *Koh et al., 2015*; *Braun et al., 2017*; *Gerhart*

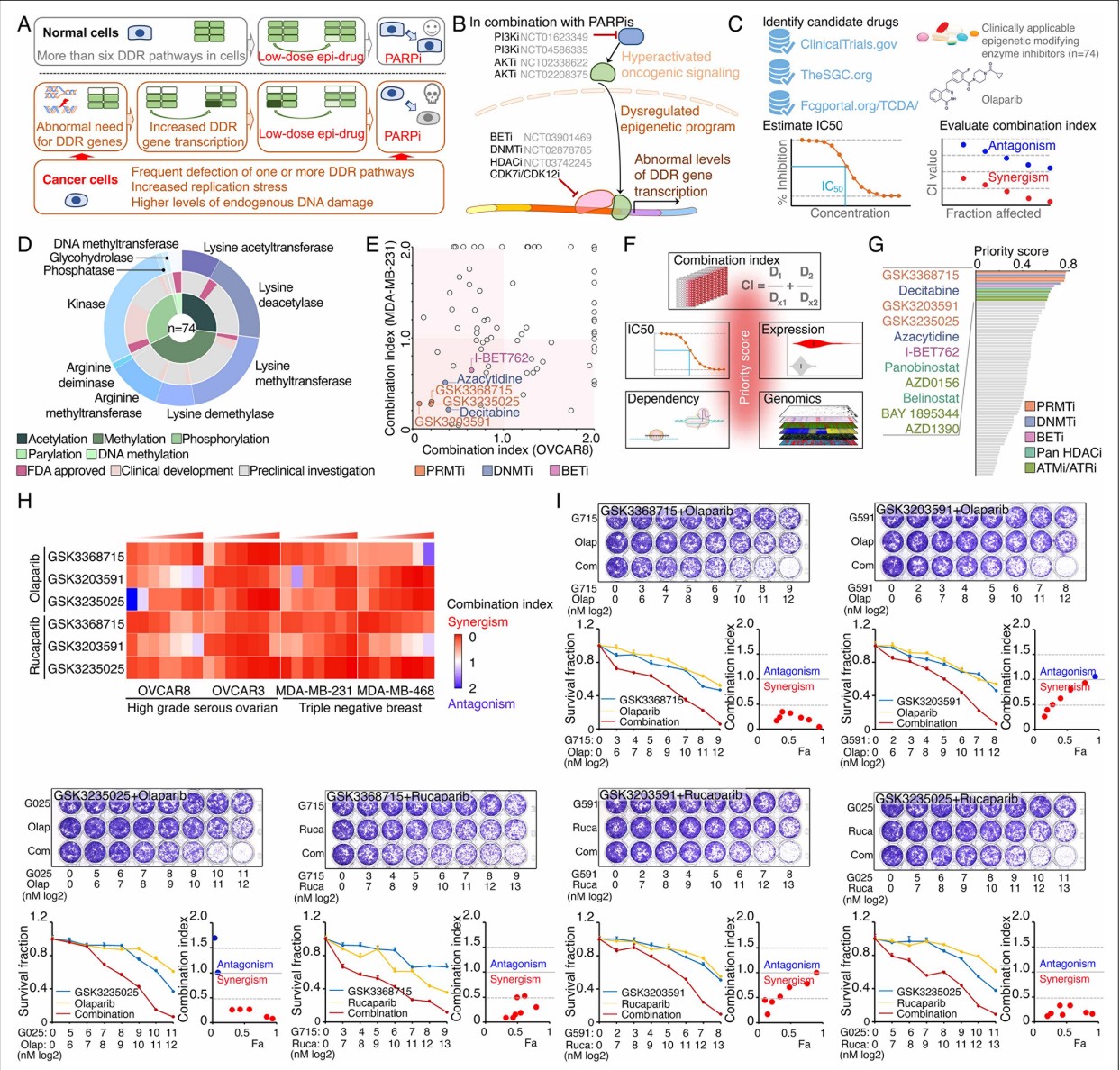

**Figure 1.** Identification of synergistic actions between PRMTis and PARPi using a drug combination screen. (**A**) Illustration of the "induced PARPi sensitivity by epigenetic modulation" strategy for treatment of cancer patients with HR-proficient tumors. Due to frequent defection of one or more DDR pathways, increased replication stress, and higher endogenous DNA damage levels, cancer cells highly rely on the DDR pathways for survival compared with normal cells. During tumorigenesis, to solve these crises, certain DDR-related genes are pathologically upregulated in cancer cells. Epigenetic mechanisms represent a major strategy used by cancer cells to maintain or restore their abnormal need for DDR gene expression, resulting in a targetable vulnerability. This 'Achilles' heel' can be targeted by epigenetic modulators. (**B**) Examples of the clinical trials based on the 'induced PARPi sensitivity by epigenetic modulation' strategy. The combinations of PARPis with the epi-drugs or small molecules that indirectly modulate DDR gene transcription were evaluated in the clinic. (**C**) Illustration of the drug combination screen approach used in this study. (**D**) The 74 epigenetic modifying enzyme inhibitors in the panel that was used for the drug combination screen were classified based on their mechanisms of action and target developmental levels. (**E**) The combination index (CI) values were generated from a drug combination screen between olaparib and each epigenetic inhibitor in OVCAR8 and MDA-MB-231 cell lines. The CI quantitatively depicts synergism (CI <0.83), additive effect (CI = 0.83–1.2), and antagonism (CI >1.2). (**F**) Five weighted features were collected and used to estimate the priority scores. (**G**) The epigenetic inhibitors were ranked based on their priority scores, and the top prioritized inhibitors are highlighted. Color indicates mechanism of action. (**H**) The heatmap shows average CI values between PARPis (olaparib or rucaparib) and PRMTis (GSK3368715, GSK3235025, or GSK3203591) in OVCAR8, OVCAR3, MDA-MB-231, and MDA-MB-468 cell lines. Color (red, synergism; blue, antagonism) intensity (light to dark) indicates increasing average CI values for each combination. (**I**) An example of the sensitivity of cancer cells to PARPi alone, PRMTi alone, and a PARPi and PRMTi combination. Only results from OVCAR8 cells are shown here. The results from other cell lines are provided in *Figure 1—figure supplement 1*. For each treatment, the upper panel shows crystal violet staining of a colony formation assay; the lower left panel shows a quantified survival fraction; and the lower right panel shows the CI values. Fa, fraction affected.

*Figure 1 continued on next page*

*Figure 1 continued*

The online version of this article includes the following figure supplement(s) for figure 1:

**Figure supplement 1.** The sensitivity of cancer cells to PARPi alone, PRMTi alone, and a PARPi and PRMTi combination in a panel of HGSOC and TNBC cell lines.

*et al., 2018*; *Fong et al., 2019*; *Radzisheuskaya et al., 2019*; *Sachamitr et al., 2021*; *Li et al., 1946*), DDR (*Pastore et al., 2020*; *Boisvert et al., 2005*; *Yu et al., 2009*; *Guo et al., 2010*; *He et al., 2011*; *Yu et al., 2012*; *Gurunathan et al., 2015*; *Clarke et al., 2017*; *Hamard et al., 2018*; *Hellmuth et al., 2018*; *Vadnais et al., 2018*; *Tan et al., 2019*; *Musiani et al., 2020*; *Wei et al., 2020*; *Sanchez-Bailon et al., 2021*; *Giuliani et al., 2021*; *Liu et al., 2022*), and immune responses (*Pastore et al., 2020*; *Kim et al., 2020*; *Wu et al., 2022*). By transferring a methyl group from S-adenosylmethionine, PRMTs catalyze the formation of arginine residues with monomethylarginine (MMA), asymmetrical dimethylarginine (aDMA), and symmetrical dimethylarginine (sDMA; *Blanc and Richard, 2017*; *Guccione and Richard, 2019*; *Jarrold and Davies, 2019*; *Lorton and Shechter, 2019*; *Wu et al., 2021*). Each such methylation state affects protein function and interactions with other proteins or nucleic acids in different ways, leading to distinct functional consequences. Based on catalytic activities, nine PRMTs in mammalians are divided into three groups. Type I (PRMT1/2/3/4/6/8) and type II (PRMT5/9) enzymes catalyze the formation of MMA as an intermediate before the establishment of aDMA and sDMA, respectively. PRMT7 is a type III enzyme that catalyzes only the formation of MMA. The activities of PRMT1 and PRMT5 are responsible for generating most cellular aDMA and sDMA, respectively; therefore, they serve as the major type I and type II enzymes in cells, respectively (*Jarrold and Davies, 2019*; *Tang et al., 2000*). Type I and II PRMTs share common substrates, with preferences for histones (such as H3R2 and H4R3; *Fuhrmann et al., 2015*) and proteins with RG/RGG motifs (such as RNA-binding proteins and splicing complex proteins; *Thandapani et al., 2013*). This substrate overlaps, and competitive behavior leads to the phenomenon of substrate scavenging (*Fong et al., 2019*; *Fedoriw et al., 2019*; *Gao et al., 2019*; *Dhar et al., 2013*). For example, PRMT1 and PRMT5 scavenge each other's substrates in cells (*Dhar et al., 2013*).

Recently, potent and selective PRMT inhibitors (PRMTis) have been successfully developed, and more than ten PRMTis have been rapidly advanced into early clinical trials (*Blanc and Richard, 2017*; *Guccione and Richard, 2019*; *Jarrold and Davies, 2019*; *Lorton and Shechter, 2019*; *Wu et al., 2021*). Although relatively rare genomic alterations in the genes encoding PRMTs are observed in cancer, expression levels of PRMTs are remarkably upregulated in solid tumors, and elevated PRMT expression is significantly associated with poor clinical outcome. Therefore, PRMTs have been considered as attractive drug targets in oncology. Preclinical studies suggest that PRMT-targeting therapies may benefit tumors with MTAP deletion (*Fedoriw et al., 2019*; *Kryukov et al., 2016*; *Mavrakis et al., 2016*; *Marjon et al., 2016*; *Kalev et al., 2021*) or splicing dysfunction (*Bezzi et al., 2013*; *Koh et al., 2015*; *Braun et al., 2017*; *Gerhart et al., 2018*; *Fong et al., 2019*; *Radzisheuskaya et al., 2019*; *Sachamitr et al., 2021*; *Metz et al., 2020*). Although PRMTis provide a new avenue for cancer treatment, translational challenges need to be addressed for their clinical application. The PRMTi targetable 'Achilles' heels' (MTAP deletion and splicing dysfunction) are only observed in a small fraction of cancers, which restricts the application of a PRMTi as a clinical monotherapy. Therefore, the development of a novel treatment strategy to expand the potential clinical applications is a key challenge for PRMTis in oncology. Most importantly, because recent clinical trial results indicated that PRMTi activity alone may be insufficient to manage patients (*Siu et al., 2019*; *Watts et al., 2019*; *Villar et al., 2020*; *McKean et al., 2021*; *Falchook et al., 2021*; *Patel et al., 2021*; *Haque et al., 2021*), researchers have had to "re-think" how to administer these drugs to patients and develop the second generation of inhibitors. Therefore, combination approaches of low-dose PRMTi with other existing therapeutic drugs urgently need to be designed and evaluated in the clinic. Excitingly, recent studies have revealed promising synergistic effects between PRMTis and chemotherapy, as well as DDR targeting drugs (*Dominici et al., 2021*; *O'Brien et al., 2023*; *Carter et al., 2023*; *Li et al., 2023*).

## Results

### Identification of synergistic actions between PRMTis and PARPi using a drug combination screen

To expand clinical applications and identify more efficient combinations for the 'induced PARPi sensitivity by epigenetic modulation' strategy, we performed a drug screen by combining PARPi with a set of epigenetic modifying enzyme inhibitors, which contain 74 well-characterized epigenetic modulators that target five major classes of epigenetic enzymatic actions, histone acetylation, histone methylation, histone phosphorylation (only kinases and phosphatases that modify histone were included), histone de-ADP-ribosylation, and DNA methylation. Among these inhibitors, 7 are FDA-approved drugs in oncology, and 14 and 54 are in clinical trials and preclinical development, respectively (*Figure 1C and D* and *Supplementary file 1A*). The BET inhibitor I-BET762, a previously identified epigenetic modulator that synergistically acts with PARPi (*Yang et al., 2017*), was used as a positive control. Two cancer cell lines that harbor wild-type BRCA1/2 and modestly respond to PARPi (OVCAR8, HGSOC; and MDA-MB-231, TNBC) were used for screening. After the IC50s of each drug were estimated in these two lines, five gradient doses (serially constant twofold ratio dilutions) were used for the drug combination screen. The combination index (CI) value of olaparib with each epigenetic inhibitor was calculated to evaluate the synergy of the combinations. In summary, synergistic effects with olaparib (medium CI value <0.83) were identified in 33 and 21 epigenetic inhibitors in OVCAR8 and MDA-MB-231, respectively, including 10 that were shared by the two lines (*Figure 1E* and *Supplementary file 1A*). Next, to prioritize the clinical application potentials, we estimated a priority score for each combination by considering the CI and four additional features of each epigenetic inhibitor (single agent treatment efficacy, cancer dependency defined by genetic screen from the DepMap project, and expression dysregulation and genomic alterations from the TCGA project, *Figure 1F*). As expected, candidates previously identified by us (*Yang et al., 2017*) and other groups, such as BETi (*Yang et al., 2017*; *Karakashev et al., 2017*; *Sun et al., 2018*), DNMTi (*Muvarak et al., 2016*; *Abbotts et al., 2019*), and HDACi (*Adimoolam et al., 2007*; *Konstantinopoulos et al., 2014*), were ranked at the top of the list (*Figure 1G* and *Supplementary file 1A*). Notably, a group of PRMT inhibitors (type I PRMTs targeting GSK3368715, and PRMT5 targeting GSK3203591 and GSK3235025) exhibited high priority scores in our analysis. This observation was further confirmed in an expanded panel of HGSOC/TNBC cell lines that were treated with each of these PRMTis in combination with two FDA-approved PARPis, olaparib and rucaparib, independently (*Figure 1H, I* and *Figure 1—figure supplement 1*).

### PRMT1, PRMT4/CARM1, and PRMT5 are potential therapeutic targets in oncology

To comprehensively characterize PRMTs in cancers, we analyzed the expressions and genomic alterations of all the PRMT family members across the TCGA cohort from 33 cancer types (*Supplementary file 1B*). An RNA-sequencing (RNA-seq) analysis indicated that although all the PRMT family members were detectable in cancers, five PRMTs (*PRMT1/2/4/5/7*) exhibited remarkably high expression levels (average FPKM >10, *Figure 2A*, *Figure 2—figure supplement 1A* and *Supplementary file 1C*). We also characterized expressional differences between tumors and their corresponding normal adjacent tissues. A meta-analysis across 21 cancer types, which had sufficient numbers of normal controls (n≥3), revealed that the mRNA levels of most PRMTs, except for *PRMT2/9*, were significantly upregulated in tumor specimens compared with corresponding controls (*Figure 2B*, *Figure 2—figure supplement 1B*). This observation was further validated at protein levels in an independent sample cohort from the CPTAC project as assessed by a mass spectrometry analysis (*Figure 2C*, *Figure 2—figure supplement 1C*). Most importantly, a meta-analysis across 33 cancer types from the TCGA cohort demonstrated that higher expressional levels of *PRMT1/3/4/5* were positively correlated with shorter patient survival times (*Figure 2D*). Next, we analyzed recurrent genomic alterations (focal somatic copy number alteration, recurrent mutation, and recurrent transcript fusion) of the PRMT family across the TCGA cohort, and no significantly recurrent events were identified at either the individual or pan-cancer levels (*Supplementary file 1D*). Finally, cancer cell-growth dependencies of PRMTs were analyzed in a large-scale CRISPR screening dataset generated by the DepMap project. Consistent with their increased expression levels and associations with poor clinical outcomes, *PRMT1/4/5* showed cancer cell-growth dependencies (*Figure 2E* and *Supplementary file 1E*). The analyses of these five features

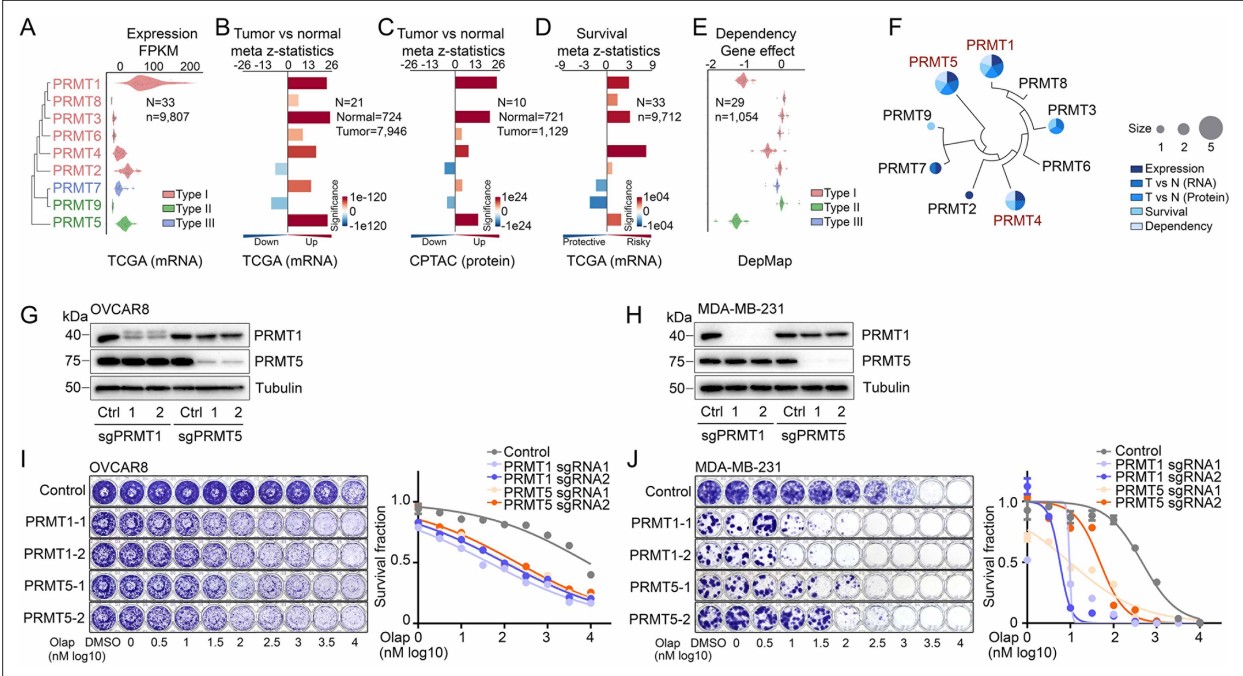

**Figure 2.** PRMT1, PRMT4/CARM1, and PRMT5 are potential therapeutic targets in oncology. (**A**) mRNA expression levels of the PRMT family across the TCGA tumor samples. Each spot represents the average mRNA expression level of a cancer type (n=33). (**B** and **C**) Meta z-statistics of mRNA (**B**) and protein (**C**) expression levels of the PRMT family in tumors compared with corresponding controls in the TCGA (**B**) or CPTAC (**C**) cohorts. Red and blue indicate up- and down-regulation, respectively. (**D**) Meta z-statistics of the association between mRNA expression and patient outcome for the PRMT family across the TCGA cohort. Red and blue indicate poor and good outcomes, respectively. (**E**) The dependency effects of the PRMT family based on the CRISPR screen from the DepMap project. Each spot represents the average gene effect of a cancer type (n=29). A lower effect score indicates that a gene is more likely to be dependent in cancer cells. (**F**) Summary of the above five features for each gene in the PRMT family. (**G** and **H**) Western blot analyses of PRMT1 and PRMT5 expression in OVCAR8 (**G**) and MDA-MB-231 (**H**) cells in which *PRMT1* and *PRMT5* were independently knocked out using lentiviral CRISPR/Cas9. (**I** and **J**) Sensitivity of olaparib in OVCAR8 (**I**) or MDA-MB-231 (**J**) cells in which *PRMT1* and *PRMT5* were independently knocked out using lentiviral CRISPR/Cas9. Left panel: crystal violet staining of a colony formation assay; right panel: survival fraction quantified from the left panel.

The online version of this article includes the following source data and figure supplement(s) for figure 2:

**Source data 1.** PDF file containing original western blots for *Figure 2G and H*, indicating the relevant bands.

**Source data 2.** Original file for western blots displayed in *Figure 2G and H*.

**Figure supplement 1.** Expression levels of the PRMTs across the TCGA tumor specimens.

indicated that two type I PRMTs (PRMT1 and PRMT4/CARM1) and one type II PRMT (PRMT5) may serve as potential therapeutic targets in oncology (*Figure 2F*). Because a PRMT4-specific inhibitor, CARM1-IN-1, did not show significantly synergistic effects with PARPi (*Figure 1B* and *Supplementary file 1A*), we hypothesized that the synergistic function of type 1 PRMT inhibitor GSK3368715 may be mediated by inhibiting PRMT1, which is the predominant enzyme among type I PRMTs (responsible for >90% activity; *Tang et al., 2000*). Consequently, we used CRISPR/Cas9 as a genetic tool to specifically knock out *PRMT1* and *PRMT5* independently in OVCAR8 and MDA-MB-231 cells (*Figure 2G and H*). Colony formation assays demonstrated that an acute knockout of *PRMT1* or *PRMT5* significantly increased the sensitivity levels of cancer cells to PARPi treatment, thereby phenocopying the results obtained using chemical PRMT inhibitors (*Figure 2I and J*).

## PRMT inhibition enhances PARPi treatment-induced DNA damage in cancer cells

To determine whether PRMTis influence DNA damage responses induced by PARPi treatments, a comet assay was used to measure DNA damage in a panel of HR-proficient cancer cells that had been treated with olaparib alone or in combination with PRMTis. Although the olaparib treatment alone induced DNA damage in all four HR-proficient cancer cell lines, the extent of the DNA

damage increased significantly when cells were treated with a combination of PRMTis (GSK3368715 or GSK3235025) and olaparib (*Figure 3A, B*, *Figure 3—figure supplement 1A*). This observation was further confirmed by a western blot assay to detect the DNA double-strand break marker, the phosphorylation of H2AX (γH2AX [S139]), among different treatment conditions (*Figure 3C*). Consistent with the enhanced DNA damage, the combination treatment significantly induced apoptosis, as measured by caspase-3/7 activity (*Figure 3D*), compared with the olaparib treatment alone. To exclude the potential off-target effects of PRMT chemical inhibitors, *PRMT1* and *PRMT5* were knocked out independently using CRISPR/Cas9 in OVCAR8 and MDA-MB-231 cells (*Figure 2G*). The specific knockout of *PRMT1* or *PRMT5* significantly enhanced the DNA damage induced by the PARPi treatment, thereby increasing apoptosis (*Figure 3E-G*, *Figure 3—figure supplement 1B*). These observations suggested that the increased expressions of *PRMT1* and *PRMT5* in cancer may prevent cell death by enhancing DNA damage repair when tumor cells are exposed to endogenous or exogenous DNA damage-causing conditions. Consequently, we analyzed the correlations between the expression levels of *PRMT1/5* and '50 hallmark' molecular signatures (*Liberzon et al., 2015*) in a large collection of cancer cell lines from the DepMap project (n=1200). Consistently, 'DNA repair' was a significant signature that positively corrected with both *PRMT1* and *PRMT5* expression levels in cancer cell lines (*Figure 3H*). Similar results were found in the primary tumor specimens across the TCGA cohort at individual cancer type and pan-cancer levels (*Figure 3I and J* and *Supplementary file 1F*). Supporting previous reports that the Myc-PRMT loop plays critical roles during tumorigenesis, 'Myc-targets' signatures were the most significant signatures correlated with the expression of *PRMT1/5* (*Figure 3H–J*).

## PRMTs maintain expression of the genes associated with DDR and BRCAness

Given that PRMTs act as key epigenetic regulators through the modification of histone methylation, we investigated gene expression changes induced by PRMTi treatments. An RNA-seq analysis revealed that GSK3368715 and GSK3235025 treatments reduced protein-coding gene expression in MDA-MB-231 cells by approximately 12.7% and 6.9%, respectively (*Figure 4A*). Consistent with correlation analyses in cancer cell lines and primary tumors (*Figure 3H–J*), a gene set enrichment analysis found that the downregulated genes in both the GSK3368715 and GSK3235025 treatment groups were significantly enriched in the DDR-associated pathways compared with controls (*Figure 4B*). The epigenetically repressed of certain DDR genes induces the BRCAness phenotype, sensitizing cancer cells to PARPi treatments. Therefore, we collected the genes in the homologous recombination (HR) and Fanconi anemia (FA) pathways (two major DDR pathways affecting PARPi sensitivity; *Knijnenburg et al., 2018*; *Wood et al., 2001*), as well as the genes that were previously reported to be associated with PARPi sensitivity (*Lord and Ashworth, 2016*). A total of 247 genes potentially implicated in PARP inhibitor sensitivity (i.e. BRCAness genes) were identified in a whole genome (*Figure 4C* and *Supplementary file 1G*). As expected, many were downregulated by GSK3368715 (n=29) and GSK3235025 (n=31) treatments, with nine being significantly repressed by both of the PRMTis (*Figure 4A* and *Supplementary file 1H*). This result was further validated by a qRT-PCR analysis (*Figure 4D*). Beyond epigenetic regulation, PRMTs play functional roles in RNA splicing through the methylation of RG/RGG motif-containing proteins in the splicing complexes. Consequently, we analyzed global RNA splicing changes between PRMTi treatments and controls. Consistent with previous reports, both the GSK3368715 and GSK3235025 treatments affected RNA splicing genome wide (*Figure 4E*). However, genes undergoing splicing changes were not significantly enriched in most DDR pathways (except for base excision repair in the GSK3235025 treatment; *Figure 4F*), although splicing alterations were identified in certain BRCAness genes, such as ERCC1, after PRMTi treatments (*Figure 4G*). Unexpectedly, we found that a PARPi treatment alone also induced modest but significant splicing alterations compared with the control (*Figure 4E*). Combination treatments with GSK3235025, but not GSK3368715, significantly further enhanced splicing alterations compared with an olaparib or GSK3235025 treatment alone (*Figure 4E*). Thus, both PRMT1i and PRMT5i treatments repressed mRNA expression of a large number of BRCAness genes in cancer cells. Although some genes were inhibited by both inhibitors, PRMT1i and PRMT5i repressed distinct BRCAness genes. Splicing alterations in certain BRCAness genes were identified after a PRMTi treatment, and the PARPi-treatment-induced splicing susceptibility may also serve as a mechanism for the synergistic action of PRMTi and PARPi.

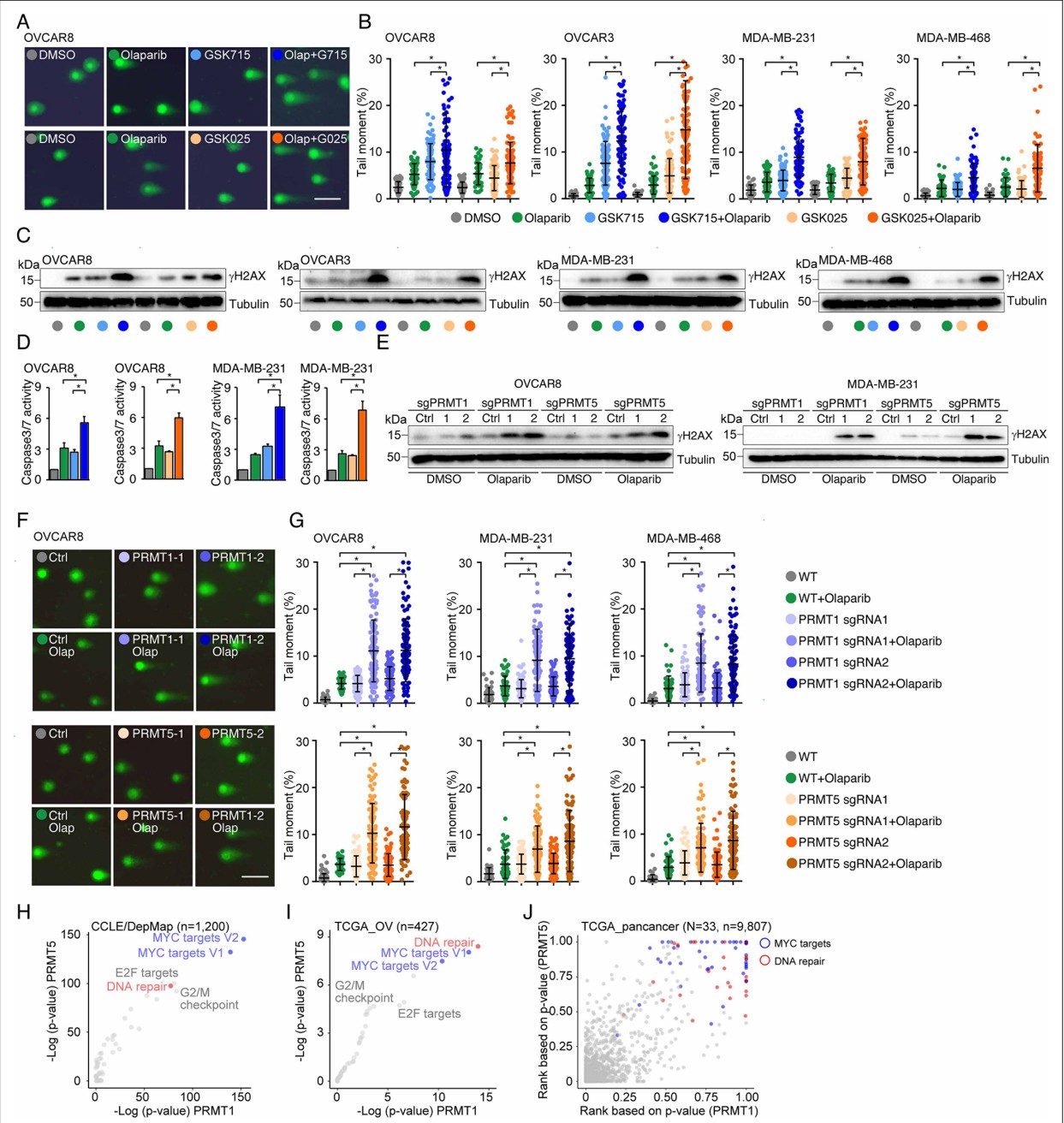

**Figure 3.** PRMT inhibition enhances PARPi treatment-induced DNA damage in cancer cells. (**A**) A comet assay was used to measure DNA damage in cancer cells treated with DMSO, olaparib (5 µM, 96 hr), GSK3368715 (5 µM, 96 hr), GSK3235025 (5 µM, 96 hr), or combinations (representative images from OVCAR8). Scale bars, 10 µm. (**B**) The extent of DNA damage was quantified using the tail moment in comet assays of OVCAR8, OVCAR3, MDA-MB-231, and MDA-MB-468 cell lines treated with DMSO, olaparib, GSK3368715, GSK3235025, or combinations. Data are presented as means ± SDs, *p<0.05 determined by two-tailed Student's t tests. (**C**) Western blot analyses of γH2AX in cancer cells treated with DMSO, olaparib, GSK3368715, GSK3235025, or combinations. (**D**) Caspase-3/7 activity was measured using a caspase-Glo 3/7 assay in OVCAR8 and MDA-MB-231 cells treated with DMSO, olaparib, GSK3368715, GSK3235025, or combinations. Data are presented as means ± SDs, n=3 biological replicates, *p<0.05 determined by two-tailed Student's t tests. (**E**) Western blot analyses of γH2AX in OVCAR8 and MDA-MB-231 cells in which *PRMT1* and *PRMT5* were independently knocked out using lentiviral CRISPR/Cas9. (**F**) A comet assay was used to measure olaparib treatment-induced DNA damage in cells in which *PRMT1* and *PRMT5* were independently knocked out using lentiviral CRISPR/Cas9 (representative images from OVCAR8). Scale bars, 10 µm. (**G**) The extent of olaparib treatment-induced DNA damage was quantified using the tail moment in comet assays of OVCAR8, MDA-MB-231, and MDA-MB-468 cell lines in which *PRMT1* and *PRMT5* were independently knocked out using lentiviral CRISPR/Cas9. Data are presented as means ± SDs, *p<0.05 determined by two-tailed Student's t tests. (**H**) and (**I**) Correlations between the expression levels of *PRMT1/PRMT5* and '50 hallmark' molecular signatures in the cancer cell lines from the DepMap (**H**) and primary tumor specimens from the TCGA (ovarian cancer was represented as an example, **I**) cohort. (**J**) Correlations

*Figure 3 continued on next page*

*Figure 3 continued*

between the expression levels of *PRMT1/PRMT5* and '50 hallmark' molecular signatures across primary tumor specimens from the TCGA cohort. For each cancer type, the correlations between *PRMT1/5* expression and '50 hallmark' molecular signatures were ranked 0–1 based on the p-values, with 1 indicating the most significant signature. Red and blue circles indicate DNA repair and Myc signatures, respectively.

The online version of this article includes the following source data and figure supplement(s) for figure 3:

**Source data 1.** PDF file containing original western blots for *Figure 3C and E*, indicating the relevant bands and treatments.

**Source data 2.** Original file for western blots displayed in *Figure 3C and E*.

**Figure supplement 1.** Comet assay was used to measure DNA damage in cancer cell lines in which PRMTs were inhibited by chemical compounds or genetic approach.

## Repression of ERCC1 is a common mechanism by which PRMTis sensitize cells to PARPi treatment

To evaluate whether the BRCAness genes, which were repressed by PRMTis, play functional roles in synergistic action between PRMTis and PARPi, a customer-designed siRNA screen was performed in MDA-MB-231 cells (*Figure 5A*), and an impact score for each candidate was estimated based on (1) expressional repression effects of PRMTis and (2) functional effects on PARPi sensitivity (*Figure 5B*). Among the candidates examined, *ERCC1* had the highest impact score and was repressed by both GSK3368715 and GSK3235025 treatments (*Figure 5C*). In lung cancer, knocking out *ERCC1* sensitizes cells to PARPi treatments (*Postel-Vinay et al., 2013*), and its expression serves as a predictive biomarker for the platinum response (*Friboulet et al., 2013*). Therefore, we used *ERCC1* as an example to evaluate whether transcriptional repression can serve as a mechanism. We found that, similar to PRMTs, *ERCC1* expression was significantly upregulated across cancer samples compared

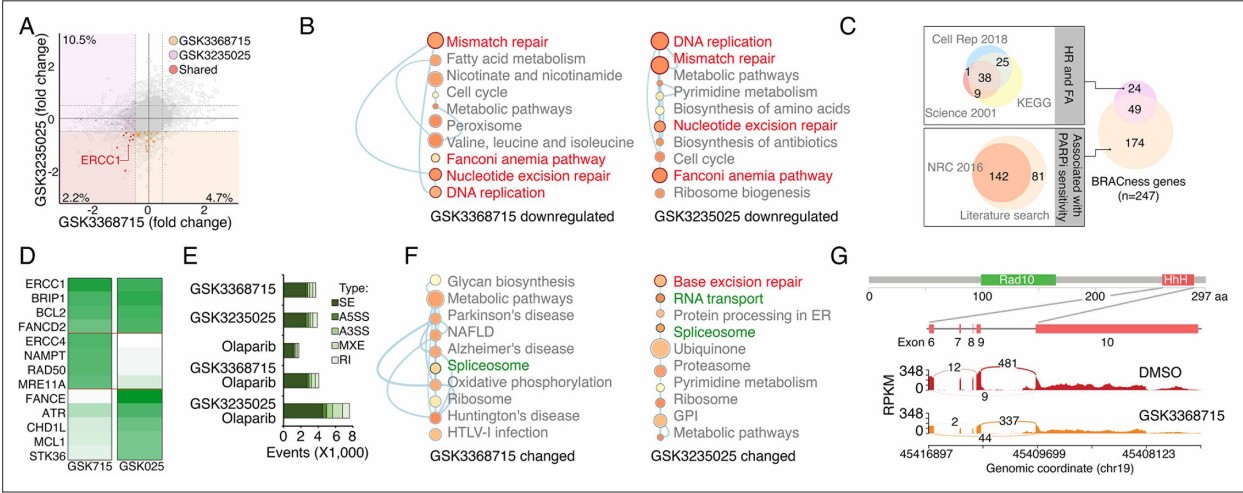

**Figure 4.** PRMTis maintain the expression of genes associated with DNA damage repair (DDR) and BRCAness. (**A**) The dot plot shows the gene expression changes measured by RNA-seq in MDA-MB-231 cells. The BRCAness genes that were repressed by GSK3368715 and GSK3235025 treatments are highlighted with color. (**B**) The top 10 KEGG pathways in which genes downregulated by GSK3368715 (left) and GSK3235025 (right) treatments were significantly enriched. Circle size indicates the fold enrichment. Lines indicate numbers of shared genes between pathways. The DDR pathways are highlighted in red. (**C**) Illustration BRCAness gene collection method. First, the genes in two major DDR pathways associated with PARPi sensitivity (HR and FA) were collected from three independent databases. In total, 73 genes annotated by at least two databases were defined as HR/FA genes. Second, the genes associated with PARPi sensitivity were retrieved from a Nature Reviews paper (n=142). An additional 81 genes were identified through a manual literature search using PubMed. In total, 223 genes were defined as PARPi sensitivity-associated genes. Taken together, 247 genes were defined as 'BRCAness genes' in this study. (**D**) The heatmap shows qRT-PCR validation of a group of BRCAness genes. Green color intensity indicates fold changes. (**E**) The abnormal splicing event numbers induced by different treatments compared with the control treatment (DMSO) in MDA-MB-231 cells. Types of abnormal splicing events are indicated by different colors. (**F**) The top 10 KEGG pathways in which abnormally spliced genes after GSK3368715 (left) and GSK3235025 (right) treatments were significantly enriched. Circle size indicates the fold enrichment. Lines indicate numbers of shared genes between two pathways. (**G**) An abnormal splicing gene (*ERCC1*). Upper panel: the structure of *ERCC1*. The alternative splicing event was identified in exons 6–10, which encodes the helix-hairpin-helix (HhH) domain of *ERCC1*. Lower panel: Sashimi plots showing the exon truncation events of *ERCC1* as detected by RNA-seq in MAD-MB-231 cells treated with DMSO and GSK3368715 independently. The X-axes indicate genomic locations, and the Y-axes indicate RNA-seq coverage. The numbers of junction reads are shown in the splice junctions (curved lines).

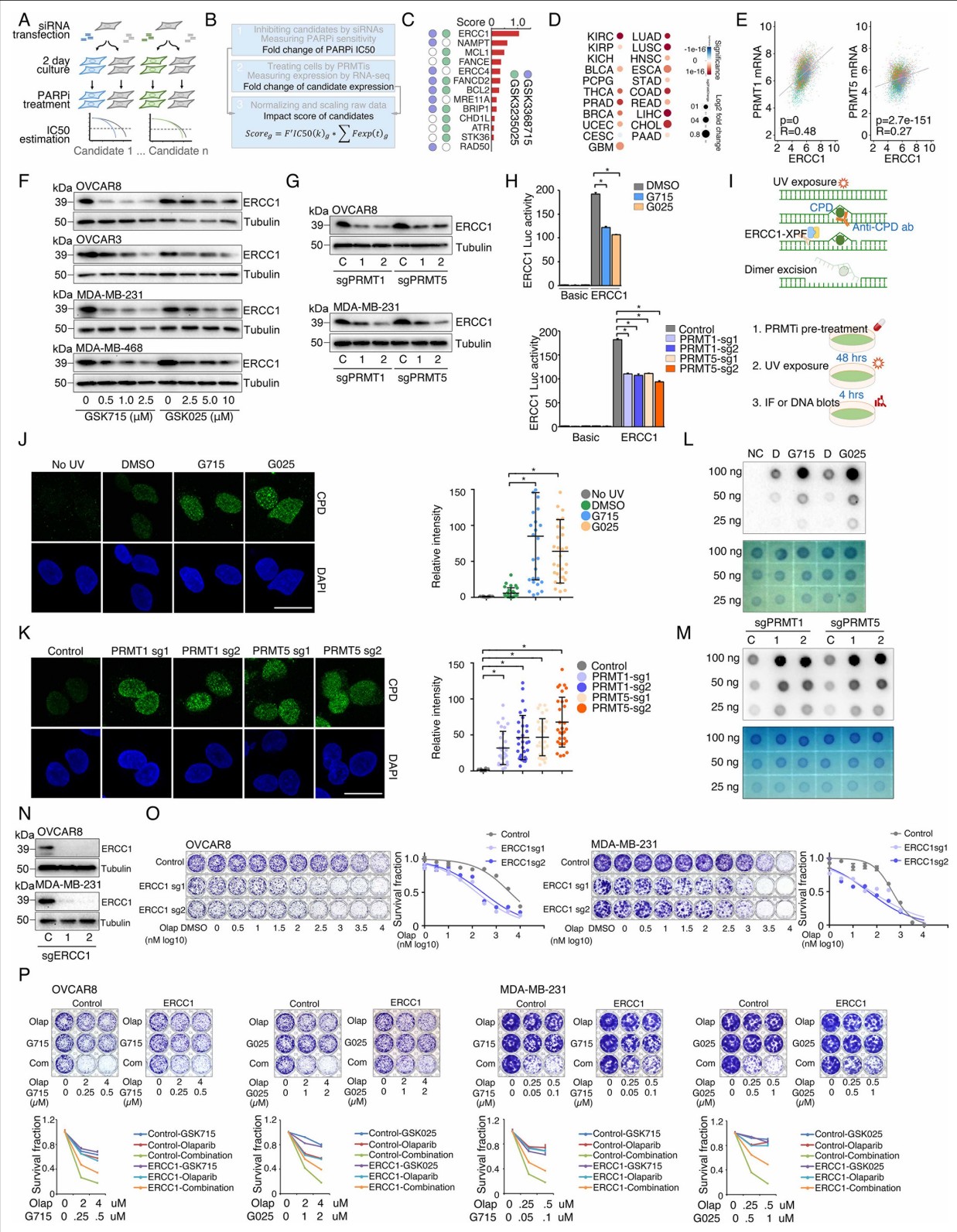

**Figure 5.** ERCC1 repression is a common mechanism by which PRMTis sensitize cells to PARPi treatment. (**A**) A siRNA screen was performed to evaluate the effects of knocking down the candidate genes on PARPi sensitivity in MDA-MB-231 cells. Gray: control siRNA transfection. (**B**) Schematic of the impact score estimation for each candidate gene based on RNA-seq and a siRNA screen. (**C**) Summary of the impact scores of 13 examined candidates. (**D**) Expressional differences in *ERCC1* mRNA levels between tumor specimens and their corresponding controls. Red and blue indicate

*Figure 5 continued on next page*

*Figure 5 continued*

up- and down-regulation, respectively. Size: fold change, intensity: p-value. (**E**) Correlations of mRNA expression levels between *ERCC1* and *PRMT1* (left) and *PRMT5* (right) across all the TCGA tumors. Color indicates cancer type. (**F**) Western blot analysis of ERCC1 protein expression in cancer cells treated with different GSK3368715 and GSK3235025 concentrations, independently, for 72 hr. (**G**) Western blot analysis of ERCC1 protein expression in OVCAR8 and MDA-MB-231 cells in which *PRMT1* and *PRMT5* were independently knocked out using lentiviral CRISPR/Cas9. (**H**) Luciferase reporter assay of *ERCC1* core promoter activity in HEK293T cells treated with GSK3368715 (5 μM, 48 hr) and GSK3235025 (5 μM, 48 hr) independently (upper panel), or after knocking out *PRMT1* and *PRMT5*, independently (lower panel). Statistical difference was assessed using Student's t test, *p<0.05; n=3. Error bars represent means ± SDs. (**I**) Schematic of UV-induced cyclobutane pyrimidine dimers (CPD) detection by antibody (upper panel) and the experimental schedule (lower panel). (**J**) Representative images (left) and quantitative results (right) of UV-induced CPD formation in MDA-MB-231 cells treated independently with DMSO, GSK3368715 (2.5 μM, 96 hr), and GSK3235025 (5 μM, 96 hr). Cells not subjected to UV treatment were used as the negative control. Scale bars, 10 μm. Statistical analysis by Student's t test, *p<0.05; n=3. Error bars represent means ± SD. (**K**) Representative images (left) and quantitative results (right) of UV-induced CPD formation in MDA-MB-231 cells in which *PRMT1* and *PRMT5* were independently knocked out using lentiviral CRISPR/Cas9. Scale bars, 10 μm. Statistical differences were assessed using Student's t test, *p<0.05; n=3. Error bars represent means ± SDs. (**L**) Dot blots of CPD levels in MDA-MB-231 cells treated independently with DMSO, GSK3368715 (5 μM, 96 hr), and GSK3235025 (5 μM, 96 hr). Cells not subjected to UV treatment were used as the negative control. Methylene blue staining was used as the loading control. (**M**) Dot blots of CPD levels in MDA-MB-231 cells in which *PRMT1* and *PRMT5* were independently knocked out using lentiviral CRISPR/Cas9. Methylene blue staining was used as the loading control. (**N**) Western blot analysis of ERCC1 protein expression in OVCAR8 and MDA-MB-231 cells in which *PRMT1* and *PRMT5* were independently knocked out using lentiviral CRISPR/Cas9. (**O**) Sensitivity of OVCAR8 and MDA-MB-231 cells to olaparib. Left panel: crystal violet staining of a colony formation assay; right panel: survival fraction quantified from the upper panel. (**P**) Sensitivity of OVCAR8 and MDA-MB-231 cells, which overexpressed ERCC1 owing to a lentiviral infection, to olaparib, GSK3368715, GSK3235025, and combinations. Upper panel: crystal violet staining of a colony formation assay; lower panel: the survival fraction quantified from the upper panel.

The online version of this article includes the following source data and figure supplement(s) for figure 5:

**Source data 1.** PDF file containing original western blots for *Figure 5F, G and N*, indicating the relevant bands and treatments.

**Source data 2.** Original file for western blots displayed in *Figure 5F, G and N*.

**Figure supplement 1.** Repression of ERCC1 is a common mechanism by which PRMTis sensitize cells to PARPi treatment.

**Figure supplement 1—source data 1.** PDF file containing original western blots for *Figure 5—figure supplement 1A and B*, indicating the relevant bands and treatments.

**Figure supplement 1—source data 2.** Original file for western blots displayed in *Figure 5—figure supplement 1A and B*.

**Figure supplement 2.** PRMT inhibition activates intrinsic innate immune pathways in cancer cells.

**Figure supplement 2—source data 1.** PDF file containing original western blots for *Figure 5—figure supplement 2E and F*, indicating the relevant bands and treatments.

**Figure supplement 2—source data 2.** Original file for western blots displayed in *Figure 5—figure supplement 2E and F*.

with their corresponding control specimens (*Figure 5D*). Consistent with the association between *PRMT1/5* and the DDR signature (*Figure 3*), significant positive correlations between *PRMT1/5* and *ERCC1* mRNA expression levels were observed in primary tumor specimens from the TCGA cohort (*Figure 5E*). These results suggest that elevated PRMT expression in cancer may serve as a mechanism to epigenetically maintain ERCC1 expression, which is critical to the survival of cancer cells that are subjected to high levels of endogenous DNA damage and increased replication-related stress. Using western blots, we demonstrated that PRMTi treatments repressed ERCC1 expression at the protein level in a dose-dependent manner (*Figure 5F*). As expected, under identical treatment conditions, GSK3368715 and GSK3235025 reduced global aDMA and sDMA levels, respectively. Notably, in consistency with previous reports (*Fedoriw et al., 2019*; *Srour et al., 2019*), the GSK3368715 treatment also increased global sDMA levels (*Figure 5—figure supplement 1*), suggesting that PRMT1 and PRMT5 share a substrate (i.e. MMA), and/or their functions may be compensatory in certain cellular contexts (*Fedoriw et al., 2019*; *Srour et al., 2019*). These observations were also validated by genetically knocking out *PRMT1* and *PRMT5* independently using CRISPR/Cas9 (*Figure 5G*, *Figure 5—figure supplement 1B*). To determine whether the repression of ERCC1 by PRMTis was mediated by an epigenetic (transcriptional) mechanism, we cloned the *ERCC1* promoter region (−1500 to +120) and inserted it into a reporter vector (pGL3-basic). Both the chemical PRMTi treatment and genetic knockout of *PRMT1/5* significantly reduced *ERCC1* promoter activities in cells (*Figure 5H*). Next, we determined whether the PRMTi treatment was able to reduce the *ERCC1* function. ERCC1 and XPF form a structure-specific nuclease that is required for the removal of ultraviolet (UV) radiation-generated cyclobutane pyrimidine dimers (CPDs). Thus, measuring the rate of UV-induced CPD removal serves as a semi-quantitative assay to monitor ERCC1 activity (*Rubbi and*

*Milner, 2003*; *Xiang et al., 2017*; *Figure 5I*). The OVCAR8 cells were pretreated with GSK3368715 (2.5 μM) and GSK3235025 (5 μM) independently for 48 hr, then CPDs were induced by exposure to UV radiation (15 J/m$^2$, *Figure 5I*). Using a specific CPD antibody (Kamiya Biomedical Company, MC-062) (*Rubbi and Milner, 2003*; *Xiang et al., 2017*), immunofluorescent staining demonstrated that PRMTi treatments significantly decreased the CPD removal rates compared with controls after UV exposure (*Figure 5J*). This observation was validated by genetically knocking out *PRMT1* or *PRMT5* using CRISPR/Cas9 (*Figure 5K*). Additionally, a DNA-blotting assay confirmed the immunofluorescent staining results (*Figure 5L and M*). To further demonstrate the functional involvement of ERCC1 in the synergistic actions between PRMTis and PARPi, we knocked out *ERCC1* in cancer cell lines using CRISPR/Cas9 (*Figure 5N*). As expected, the ERCC1-XPF activity was significantly reduced (*Figure 5—figure supplement 1C*). Colony formation assays demonstrated that *ERCC1* knockout cancer cell lines were sensitized to PARPi treatments (*Figure 5O*). Finally, the forced ERCC1 expression by lentiviral infection partially reduced the synergistic actions between PRMTis and PARPi in these cell lines (*Figure 5P*). Therefore, both type I PRMT and PRMT5 inhibitor treatments significantly repressed ERCC1 expression and activity levels in cancer cells. ERCC1 serves as a common mechanism by which PRMTis sensitize cells to PARPi treatments.

## PRMT inhibition activates intrinsic innate immune pathways in cancer cells

We also analyzed the PRMTi treatment-induced genes using a gene set enrichment analysis and found that the most upregulated pathways were associated with immune responses (*Figure 5—figure supplement 2A*). Using the immune-related gene list from an immune gene annotation database (ImmPort), 516 reliably detectable genes (RPKM >1 by RNA-seq) in MDA-MB-231 cells were defined as the immune-related genes. Consistently, we found that 17.8% and 23.6% were upregulated by GSK3368715 and GSK3235025 treatments, respectively (*Figure 5—figure supplement 2B*). Compared with the upregulated non-immune-related genes, a significant number of the immune-related genes were induced by PRMT inhibition (GSK3368715, p=7.6e-9; GSK3235025, p=1.8e-18). Notably, 53 immune-related genes were upregulated by both PRMTis, and 37.7% of them were classified as innate immune-related genes by the InnateBD database (*Figure 5—figure supplement 2C*). These innate immune-related genes were associated with the major innate immune signaling pathways, such as cytokines, chemokines, and their receptors (*Figure 5—figure supplement 2D*). These results suggest that PRMTi treatments may enhance intrinsic immune reactions in tumor cells, providing a strong rationale for the application of PRMTis (or in combination with immune checkpoint inhibitors) in immuno-oncology. Notably, PARPi treatments trigger anticancer innate immune responses by increasing cytosolic DNA. Consequently, using the phosphorylation of TBK1 as a functional marker to monitor the activities of innate immune signals in tumor cells, we found that the PRMTi treatment alone induced comparable levels of p-TBK1 as the PARPi treatment, whereas the combination of PRMTi and PARPi increased p-TBK1 remarkably compared with single agent treatments (*Figure 5—figure supplement 2E*). This observation was confirmed by genetically knocking down *PRMT1* and *PRMT5* independently using CRISPR/Cas9 (*Figure 5—figure supplement 2F*). Because both PRMTi and PARPi treatments induced DNA damage in cancer cells, we also examined cytoplasmic micronuclei using PicoGreen staining, which specifically recognizes dsDNA (*Figure 5—figure supplement 2G*). Indeed, the numbers of cytoplasmic micronuclei significantly accumulated when cells were treated with PRMTis or PARPi alone, whereas the combination treatment synergistically enhanced this response (*Figure 5—figure supplement 2H*). The combination of PRMTis and PARPi may synergistically enhance anticancer immunity because cytoplasmic dsDNA triggers the cGAS-STING-TBK1 pathway (*Kwon and Bakhoum, 2020*).

## Type I PRMT and PRMT5 inhibition act synergistically to enhance PARPi sensitivity

The inhibition of type I PRMT and PRMT5 represses shared, as well as distinct, BRCAness genes (*Figure 4A*); consequently, we hypothesized that the combination of type I PRMT and PRMT5 inhibitors may act synergistically to enhance PARPi sensitivity. Thus, using ERCC1 as an example, we examined whether there was a synergistic effect on the repression of shared targets. Both chemical (PRMTis) and genetic (CRISPR) approaches demonstrated that the repression of both PRMT1 and

PRMT5 more significantly reduced *ERCC1* promoter activities than single treatments (*Figure 6A and B*). This transcriptional synergistic action was further confirmed by qRT-PCR (*Figure 6C*). More importantly, the combination treatment almost completely repressed ERCC1 protein expression in cancer cell lines (*Figure 6D*). Using combinations of inhibitors that target the same pathway or biological process not only leads to synergistic effects and overcomes resistance, it may also reduce the required dose of each drug, thereby reducing potential side effects and increasing the therapeutic window. For example, co-targeting type I PRMTs and PRMT5 using a combination of different PRMTis showed synergistic actions and promising therapeutic efficacies in preclinical models (*Fong et al., 2019*; *Fedoriw et al., 2019*; *Gao et al., 2019*). Because PARPis are well tolerated in clinical practice, we next evaluated whether de-escalation doses of PRMTis in combination with PARPi can achieve a therapeutic efficacy in a panel of cancer cells (*Figure 6E*). Decreased independent doses of GSK3368715 (125 nM) and GSK3235025 (500 nM) achieved identical synergistic effects as the original doses of these two inhibitors (*Figure 6E*). Furthermore, although extremely low doses of GSK3368715 (25 nM) and GSK3235025 (100 nM) independently showed moderate synergy with olaparib, a triple combination produced a strong synergistic action with olaparib. More importantly, when we further reduced doses of PRMTis, synergistic effects were still achieved (i.e. GSK3368715 at 12.5 nM and GSK3235025 at 12.5 nM) in the triple combination setting (*Figure 6F*). Similar results were observed for another FDA-approved PARPi, rucaparib (*Figure 6G*, *Figure 6—figure supplement 1*). Finally, we validated our in vitro findings through in vivo experiments using immunodeficient nude mouse models treated with low doses of PRMT inhibitors (*Figure 6H–K*). Consistently, olaparib in combination with GSK3368715 and GSK3235025 independently significantly reduced tumor growth in vivo compared with the olaparib treatment alone, although in this low-dose PRMTi setting, PRMTi alone showed no (GSK3235025) or modest (GSK3368715) effects on tumor growth. The triple combination achieved a stronger therapeutic effect compared with the double combinations. Additionally, consistent with previous reports (*Fedoriw et al., 2019*), combinations of GSK3368715 and GSK3235025 also showed promising therapeutic effects compared with GSK3368715 and GSK3235025 alone. Notably, GSK3368715 exhibited greater potency than both GSK3235025 and Olaparib. Treatment with GSK3368715 alone resulted in significantly stronger tumor growth inhibition compared with either GSK3235025 or Olaparib alone. This higher potency likely explains the comparable tumor suppression observed in the combination groups (GSK3368715 plus Olaparib and GSK3368715 plus GSK3235025), indicating that GSK3368715 is the primary driver of efficacy in these two-drug combination settings. Importantly, no significant side effects were observed in PRMTi and PARPi combinations and the triple combination in this low-dose PRMTi setting (*Figure 6L*), suggesting that both double and triple combinations may be tolerable in future clinical developments. Finally, to evaluate whether PRMTis reduce BRCAness gene expression in vivo, we conducted a short-term treatment experiment (*Figure 6M*). After one week of PRMTi administration, tumor tissues were collected and *ERCC1* expression was measured by qPCR. Consistently, inhibition of either Type I PRMT or PRMT5 alone significantly reduced *ERCC1* expression, while the combination of these inhibitors produced an even greater reduction (*Figure 6N*). These findings further support that downregulation of the BRCAness gene is one of the mechanisms contributing to the enhanced efficacy of combined PRMTi and PARPi treatment.

## Discussion

An 'induced PARPi sensitivity by epigenetic modulation' strategy (i.e. epigenetically impairing the genes of the DDR pathway in HR-proficient tumors to induce an HR-deficiency or other DDR defects, thereby enhancing and sensitizing tumor cells to PARPi) has been developed recently to expand PARPi clinical applications (*Konstantinopoulos et al., 2015*; *Pilié et al., 2019*; *Yang et al., 2017*). For example, our previous study has demonstrated that repression of BET activity can significantly reduce the expression levels of essential HR genes by inhibiting the transcriptional activities of their super enhancers (*Yang et al., 2017*). This preclinical discovery is being evaluated in a Phase 1b/2 clinical trial (NCT03901469) by a combination of BETi (ZEN-3694) with PARPi (talazoparib) for treatment of patients with metastatic TNBC without germline BRCA1/2 mutations (i.e. intrinsic HR-proficient patients; *Aftimos et al., 2022*). Excitingly, promising anti-cancer activity of this novel drug combination has been observed in the ongoing clinical trial. Importantly, paired tumor biopsies have shown robust target engagement (repression of *BRCA1* and *RAD51* mRNA expression) (*Aftimos et al., 2022*), consistent with our preclinical studies in xenograft tumor models (*Yang et al., 2017*). This trial

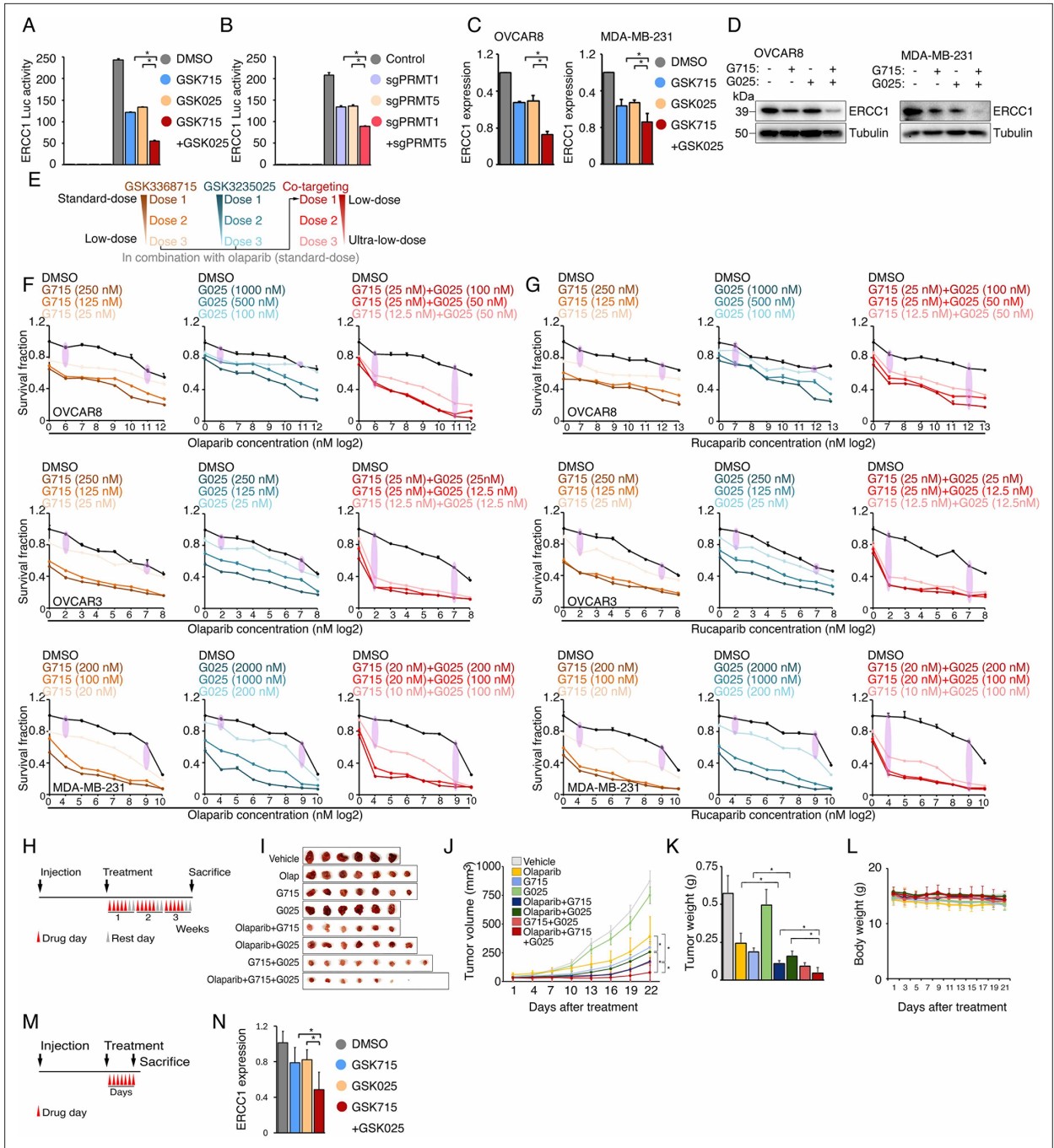

**Figure 6.** Type I PRMT and PRMT5 inhibition acts synergistically to enhance PARPi sensitivity. (**A**) Luciferase reporter assay of the *ERCC1* core promoter activity levels in HEK293T cells treated with DMSO, GSK3368715 (5 μM, 48 hr), and GSK3235025 (5 μM, 48 hr) independently, or with the combination of GSK3368715 (5 μM, 48 hr) and GSK3235025 (5 μM, 48 hr). Statistical differences were assessed using Student's t test, *p<0.05; n=3. Error bars represent means ± SDs. (**B**) Luciferase reporter assay of the *ERCC1* core promoter activity levels in HEK293T cells in which *PRMT1*, *PRMT5*, and *PRMT1/5* were independently knocked out using lentiviral CRISPR/Cas9. Statistical differences were assessed using Student's t test, *p<0.05; n=3. Error bars represent means ± SDs. (**C**) qRT-PCR analysis of *ERCC1* RNA expression in OVCAR8 and MDA-MB-231 cells treated with GSK3368715 (5 μM) and GSK3235025 (5 μM) independently for 48 hr. (**D**) Western blot analysis of ERCC1 protein expression in OVCAR8 and MDA-MB-231 cells treated with GSK3368715 (5 μM) and GSK3235025 (5 μM) independently for 72 hr. (**E**) Dose de-escalation scheme. The lowest doses of GSK3368715 and GSK3235025 independently for PRMTi and the PARPi combination were first identified. They were then used as initial doses for triple combination. The doses of GSK3368715 and GSK3235025 were further reduced until the lowest doses for each PRMTi in the triple combination were identified. (**F**) Sensitivity levels of OVCAR8 (upper panel), OVCAR3 (middle panel), and MDA-MB-231 (lower panel) cells to olaparib alone or olaparib combined with three different doses of GSK3368715 (left panel), GSK3235025 (middle panel), and GSK3368715 plus GSK3235025 (right panel). (**G**) Sensitivity levels of OVCAR8

*Figure 6 continued on next page*

*Figure 6 continued*

(upper panel), OVCAR3 (middle panel), and MDA-MB-231 (lower panel) cells to rucaparib alone or rucaparib combined with three different doses of GSK3368715 (left panel), GSK3235025 (middle panel), and GSK3368715 plus GSK3235025 (right panel). (**H**) Schematic of the MDA-MB-231 xenograft experimental design. MDA-MB-231 cells were implanted subcutaneously in immunodeficient nude mice and grown until tumors reached a size of approximately 30 mm³. Tumor-bearing mice were randomized and then received vehicle, 50 mg/kg olaparib i.p., 25 mg/kg GSK3368715 p.o., 50 mg/kg GSK3235025 p.o., or combinations of two agents or three agents, 5 days a week for 3 weeks. Caliper measurements were taken every other day from the initiation of drug treatment. (**I**) Images of tumors collected from animals receiving vehicle, olaparib, GSK3368715, GSK3235025, and combinations of two agents or three agents. (**J**) Tumor growth curve for each treatment group in immunodeficient nude mice. Statistical significance was assessed using a two-tailed Student's t test. Error bars represent means ± SDs. (**K**) Bar plot showing the mean tumor volume of each treatment group. Statistical significance was assessed using a two-tailed Student's t test. Error bars represent means ± SDs. (**L**) Changes in the body weights of different murine treatment groups. Statistical significance was assessed using a two-tailed Student's t test. Error bars represent means ± SDs. (**M**) Schematic of the MDA-MB-231 xenograft experimental design. MDA-MB-231 cells were implanted subcutaneously in immunodeficient nude mice and grown until tumors reached a size of approximately 200 mm³. Tumor-bearing mice were randomized and then received vehicle, 25 mg/kg GSK3368715 p.o., 50 mg/kg GSK3235025 p.o., or combination of two agents, for 1 week. (**N**) qRT-PCR analysis of ERCC1 RNA expression in MDA-MB-231 xenograft tumors following 1 week of treatment.

The online version of this article includes the following source data and figure supplement(s) for figure 6:

**Source data 1.** PDF file containing original western blots for *Figure 6D*, indicating the relevant bands and treatments.

**Source data 2.** Original file for western blots displayed in *Figure 6D*.

**Figure supplement 1.** PRMT1 and PRMT5 inhibition acts synergistically to enhance PARPi sensitivity in a panel of HGSOC and TNBC cell lines.

is being expanded to a Phase 2b stage to accrue additional patients with TNBC. In addition, other combination approaches (*Yang et al., 2017*; *Karakashev et al., 2017*; *Sun et al., 2018*; *Johnson et al., 2016*; *Iniguez et al., 2018*; *Shan et al., 2020*; *Muvarak et al., 2016*; *Abbotts et al., 2019*; *Baer et al., 2022*; *Adimoolam et al., 2007*; *Konstantinopoulos et al., 2014*; *Ibrahim et al., 2012*; *Juvekar et al., 2012*; *Konstantinopoulos et al., 2019*; *Batalini et al., 2022*; *Yap et al., 2020*; *Westin et al., 2021*; *Mo et al., 2016*; *Sun et al., 2017*) based on the 'induced PARPi sensitivity by epigenetic modulation' strategy also achieved promising clinical responses in intrinsic or acquired HR-proficient settings (*Baer et al., 2022*; *Konstantinopoulos et al., 2019*; *Batalini et al., 2022*; *Yap et al., 2020*; *Westin et al., 2021*; *Aftimos et al., 2022*). Most importantly, due to cancer cells being extremely sensitive to the epigenetic repression of DDR genes compared with normal cells, the clinical studies demonstrated that this strategy is well tolerable for the patients with cancer.

However, two key clinically relevant challenges still need to be addressed for future expansion of the application of this strategy in oncology: that is which clinically applicable epi-drug(s) can provide the strongest synergistic effects with PARPi? And can a BRCA-independent strategy be developed? To design and evaluate highly effective and BRCA-independent epi-drug/PARPi combination strategy, we performed a drug screen by combining olaparib, an FDA-approved PARPi, with a set of clinically applicable epi-drugs, which contain 74 well-characterized epigenetic modulators that target five major classes of epigenetic enzymatic actions. Among them, 7 are FDA-approved drugs in oncology, and 14 and 54 are in clinical trials and preclinical development, respectively. Notably, both type I PRMTi and PRMT5i exhibited high combination and clinical priority scores in our screen. This result was further confirmed by using different FDA-approved PARPis in multiple HGSOC and TNBC cell lines. Our functional studies demonstrated that PRMT inhibition significantly enhanced PARPi treatment-induced DNA damage in HR-proficient cancer cell lines. In consistency with the significant and positive correlation between the expression of PRMTs and DDR genes across primary cancer specimens, we observed that PRMT activities maintain the transcription of genes associated with DNA damage repair and 'BRCAness' in cancer cells.

The expression levels of many PRMT family members are significantly elevated in cancer cells and associated with poor clinical outcomes, strongly suggesting that they may serve as a class of novel therapeutic targets in oncology. Our large-scale genomic and functional profiles from TCGA and DepMap further confirm that PRMT1, PRMT4, and PRMT5 are potential therapeutic targets in oncology. Major pharmaceutical companies have developed more than 10 selective and potent PRMTis that are currently undergoing early-stage clinical evaluation. Development of novel treatment strategy to expand its potential clinical application is a key challenge for application of PRMTis in oncology. Given that recent clinical trials indicate that the activity of PRMTi alone may be insufficient to manage patients with cancer (*Siu et al., 2019*; *Watts et al., 2019*; *Villar et al., 2020*; *McKean*

*et al., 2021*; *Falchook et al., 2021*; *Patel et al., 2021*; *Haque et al., 2021*), even in MTAP-deletion setting, therefore, the design and evaluation of combination approaches using PRMTi with other therapeutic drugs are urgently needed. Our preclinical studies provide a strong rationale for a combination of PRMTi with PARPi for treatment of patients with HR-proficient HGSOC or TNBC. In addition, this strategy may also be expanded to other cancer types such as prostate and pancreatic cancers, in which PARPi has been approved by the FDA for clinical management.

As major histone modifiers, PRMTs control methylation of histones, thereby epigenetically regulating gene transcription and expression (*Blanc and Richard, 2017*; *Guccione and Richard, 2019*; *Jarrold and Davies, 2019*; *Lorton and Shechter, 2019*; *Wu et al., 2021*). Additionally, through protein post-translational modification, PRMTs also directly regulate non-histone protein functions. For example, PRMTs preferentially modify the splicing factors to maintain appropriate splicing process in cells. Given that the PRMT-regulome broadly influences multiple cellular processes in cancer cells, the mechanisms of action for the PRMT and PARP inhibitor combination may be highly complicated and context-dependent. For example, it has been reported that PRMTs indirectly regulate DDR gene expression through their functions on maintaining arginine methylation of the proteins in splicing complex (*Hamard et al., 2018*; *Tan et al., 2019*; *Giuliani et al., 2021*). Consistent with these studies, we also found that splicing of certain DDR genes, such as ERCC1, was impaired by PRMT inhibition. Additionally, PRMTs may also directly impact arginine methylation in certain DDR genes, thereby influencing their function in DNA damage response (*Boisvert et al., 2005*; *Yu et al., 2009*; *Guo et al., 2010*; *He et al., 2011*; *Yu et al., 2012*; *Gurunathan et al., 2015*; *Clarke et al., 2017*; *Hellmuth et al., 2018*; *Vadnais et al., 2018*; *Musiani et al., 2020*; *Sanchez-Bailon et al., 2021*). In the context of HGSOC/TNBC, our results suggest that treatment with both type 1 PRMT and PRMT5 inhibitors suppressed RNA expression of many genes potentially involved in PARP inhibitor sensitivity (i.e. BRCAness genes). Although some of them were inhibited by both inhibitors, type 1 PRMT and PRMT5 inhibitors repressed distinct BRCAness genes. Splicing alterations in certain BRCAness genes were identified after a PRMTi treatment; therefore, the PRMTi-treatment-induced splicing susceptibility may also serve as an alternative mechanism for the synergistic action of PRMTi and PARPi. However, BRCAness genes were not significantly enriched among the genes whose splicing was significantly altered by PRMTi treatment, suggesting that splicing alteration might not serve as a dominant mechanism for PRMTi/PARPi combination, at least in the HGSOC/TNBC setting. Our studies not only provided mechanistic explanations for PRMTi and PARPi combinations but also provided novel information for the clinical development of this strategy. For example, ERCC1 expression may serve as a predictive biomarker for patient selection. Notably, ERCC1 is being evaluated in the clinic as a novel biomarker for predicting platinum sensitivity in lung cancer (*Friboulet et al., 2013*; *Olaussen et al., 2006*; *Lee et al., 2017*; *Ganzinelli et al., 2021*). Interestingly, an additional band appeared after PRMT1 knockdown in OVCAR8 cells using two sequence-independent gRNAs, but not in MDA-MB-231 cells. This likely represents a low-abundance PRMT1 isoform whose expression increased upon knockdown of the major isoforms via CRISPR/Cas9. Given the tissue-specific nature of PRMT1 isoforms (*Goulet et al., 2007*), this band was absent in MDA-MB-231 cells. This observation raises an important question regarding the role of PRMT1 isoforms in regulating the DNA damage response in cancer. Furthermore, we found splicing alterations of *ERCC1* accompanied by a reduction in total *ERCC1* mRNA reads following PRMTi treatment. This decrease may also be partly due to the instability of alternatively spliced ERCC1 transcripts, which are more prone to degradation. Together with PRMTi-induced transcriptional downregulation, these splicing changes likely further reduce functional ERCC1 protein levels. This dual impact may also contribute to the enhanced cellular sensitivity to PARP inhibitors. Finally, tumor intrinsic genomic status may affect responses to PRMTi alone or combination treatments (*Blanc and Richard, 2017*; *Guccione and Richard, 2019*; *Jarrold and Davies, 2019*; *Lorton and Shechter, 2019*; *Wu et al., 2021*). Supporting previous reports found that the Myc-PRMT loop plays critical roles during tumorigenesis (*Koh et al., 2015*), and 'Myc-targets' signatures were the most significant signatures correlated with the expression of PRMT1 and PRMT5. Therefore, under MYC-hyperactivated conditions, tumors may be extremely sensitive to PRMTis or PRMTi/PARPi combination.

Co-targeting type I PRMT and PRMT5 has shown strong synergistic actions and promising therapeutic efficacies (*Fong et al., 2019*; *Fedoriw et al., 2019*; *Gao et al., 2019*). Combinations of drugs that target the same pathway (e.g. combination of dabrafenib and tremetinib) not only lead to

synergistic effects and overcome resistance, they may also reduce the required dose of each drug, decreasing side effects and increasing the therapeutic window (*Srour et al., 2019*). In a PARPi treatment setting, type I PRMT and PRMT5 regulated shared as well as distinct BRCAness genes, suggesting that co-targeting type I PRMT and PRMT5 may produce a synergistic effect. Indeed, we observed that dual treatments of type I PRMT and PRMT5 more significantly reduced ERCC1 expression compared with single treatments. Notably, in consistency with previous reports (*Fedoriw et al., 2019*; *Srour et al., 2019*), PRMT1i increased global sDMA levels, which suggested that type I PRMT and PRMT5 share a substrate (i.e. MMA) and/or their functions are compensatory. Therefore, we believe that co-targeting type I PRMT and PRMT5 not only results in a synergistic effect that enhances PARPi sensitivity, reducing the required dose of each drug, but it also serves as a strategy to overcome potential resistance due to the compensatory functions of type I PRMT and PRMT5. This concept was supported by our in vitro triple combination experiments as well as in vivo studies. Although treatment-related adverse events are manageable and reversible with treatment interruption and reduction, efficacy of PRMTis as monotherapeutic agents was modest in clinical trials (*Siu et al., 2019*; *Watts et al., 2019*; *Villar et al., 2020*; *McKean et al., 2021*; *Falchook et al., 2021*; *Patel et al., 2021*; *Haque et al., 2021*). Therefore, novel combination strategies using lower doses of PRMTis urgently need to be designed and evaluated. Our studies indicated that, because PRMT 'addiction' enables tumor cells to survive DNA damage, targeting PRMTs can lead to 'treatment-induced DNA damage sensitivity', thereby sensitizing tumor cells to PARPi (and other DNA damaging drugs) treatments. A low-dose PRMTi in combination with PARPi may represent a more tolerable strategy for future clinical applications in oncology. Finally, co-targeting type I PRMT and PRMT5 may further significantly reduce the doses of each PRMTi (i.e. ultra-low-dose), thereby decreasing side toxicity effects of PRMTis. Taken together, our studies provide a strong rationale for clinical application of a combination of PRMT and PARP inhibitors in the patients with HR-proficient ovarian or breast cancer. Our mechanistic characterization may contribute to the future clinical development of PRMTis by defining mechanism of action, identifying rationales for combination, selecting response predictive biomarkers, and guiding dose and treatment schedules.

In summary, our study identifies PRMT1 and PRMT5 as key epigenetic regulators of DNA damage repair and shows that their inhibition sensitizes HR-proficient tumors to PARP inhibitors by repressing transcription and altering splicing of BRCAness genes. Distinct from prior strategies, dual inhibition of type I PRMT and PRMT5 exhibits strong synergy, allowing for lower-dose combination treatments that may reduce toxicity. Our findings also nominate ERCC1 as a potential predictive biomarker and suggest that MYC-driven tumors may be particularly responsive to this approach. Collectively, these results offer a mechanistic rationale and translational framework to broaden the clinical application of PARP inhibitors.

## Materials and methods

**Key resources table**

| Reagent type (species) or resource | Designation | Source or reference | Identifiers | Additional information |
|---|---|---|---|---|
| Cell line (*Homo sapiens*) | OVCAR8 | NCI-DTP | NCI-60 human cancer cell line panel; RRID:CVCL_1629 | |
| Cell line (*Homo sapiens*) | OVCAR3 | ATCC | Cat: #HTB-161; RRID:CVCL_DH37 | |
| Cell line (*Homo sapiens*) | MDA-MB-231 | ATCC | Cat: #CRM-HTB-26; RRID:CVCL_0062 | |
| Cell line (*Homo sapiens*) | MDA-MB-468 | ATCC | Cat: #HTB-132; RRID:CVCL_0419 | |
| Cell line (*Homo sapiens*) | HEK293T | ATCC | Cat: #CRL-3216; RRID:CVCL_0063 | |
| Antibody | Anti-PRMT1 (Rabbit polyclonal) | Cell Signaling Technology | Cat: #2449 S; RRID:AB_2237696 | WB (1:1000) |

*Continued on next page*

*Continued*

| Reagent type (species) or resource | Designation | Source or reference | Identifiers | Additional information |
|---|---|---|---|---|
| Antibody | Anti-PRMT5 (Rabbit monoclonal) | Cell Signaling Technology | Clone:D5P2T; Cat: #79998 S; RRID:AB_2799945 | WB (1:1000) |
| Antibody | Anti-aDMA (Rabbit monoclonal) | Cell Signaling Technology | Cat: #13522 S; RRID:AB_2665370 | WB (1:1000) |
| Antibody | Anti-sDMA (Rabbit monoclonal) | Cell Signaling Technology | Cat: #13222 S; RRID:AB_2714013 | WB (1:1000) |
| Antibody | Anti-ERCC1 (Rabbit polyclonal) | Cell Signaling Technology | Cat: #3885 S; RRID:AB_2100142 | WB (1:1000) |
| Antibody | Anti-phospho-H2AX (S139) (Mouse monoclonal) | Sigma-Aldrich | Clone:JBW301; Cat: #05–636; RRID:AB_309864 | WB (1:1000) |
| Antibody | Anti-phospho-TBK1/NAK (Rabbit monoclonal) | Cell Signaling Technology | Clone:D52C2; Cat: #5483 S; RRID:AB_10693472 | WB (1:1000) |
| Antibody | Anti-β-Tubulin (Rabbit monoclonal) | Cell Signaling Technology | Clone:9 F3 Cat: #2128; RRID:AB_823664 | WB (1:1000) |
| Antibody | Anti-Cyclobutane Pyrimidine Dimers (CPDs) (Mouse monoclonal) | Cosmo Bio | Clone:TDM-2; Cat: #CAC-NM-DND-001; RRID:AB_1962813 | IF (1:1000) |
| Recombinant DNA reagent | LentiCRISPRv2 (plasmid) | Addgene | Cat:#98290 | |
| Recombinant DNA reagent | LentiCRISPRv2-PRMT1-sgRNA1 (plasmid) | This paper | | PRMT1 targeting CRISPR plasmid；Guide:GTGGATGCCAAAGTGTGCGT |
| Recombinant DNA reagent | LentiCRISPRv2-PRMT1-sgRNA2 (plasmid) | This paper | | PRMT1 targeting CRISPR plasmid；Guide:CCTTGGCAGCAAACATGCAG |
| Recombinant DNA reagent | LentiCRISPRv2- PRMT5-sgRNA1 (plasmid) | This paper | | PRMT5 targeting CRISPR plasmid；Guide:ATGAACTCCCTCTTGAAACG |
| Recombinant DNA reagent | LentiCRISPRv2- PRMT5-sgRNA2 (plasmid) | This paper | | PRMT5 targeting CRISPR plasmid；Guide:GATTCGTCCAGACTCAAAAG |
| Recombinant DNA reagent | LentiCRISPRv2- ERCC1-sgRNA1 (plasmid) | This paper | | ERCC1 targeting CRISPR plasmid；Guide:ATTTGTGATACCCCTCGACG |
| Recombinant DNA reagent | LentiCRISPRv2- ERCC1-sgRNA2 (plasmid) | This paper | | ERCC1 targeting CRISPR plasmid；Guide:ATATTCGGCGTAGGTCTGAG |
| Recombinant DNA reagent | CD513B (plasmid) | System Biosciences | Cat: #CD513B | |
| Recombinant DNA reagent | CD513B-human ERCC1 (plasmid) | This paper | | ERCC1 over-expression in CD513B |
| Recombinant DNA reagent | pGL3 Basic Vector (plasmid) | Addgene | Cat: #212936 | |
| Recombinant DNA reagent | pGL3 Basic-ERCC1 promoter Vector (plasmid) | This paper | | ERCC1 promoter version of pGL3 Basic |
| Commercial assay or kit | Dual-Luciferase Reporter Assay System | Promega | Cat: #E1910 | |
| Chemical compound, drug | Crystal violet | Millipore Sigma | Cat: #32675 | Colony formation assay (0.5%) |
| Chemical compound, drug | SYBR Gold Nucleic Acid Gel Stain | Thermo Fisher Scientific | Cat: #S11494 | Comet assay (1:10,000) |

*Continued on next page*

*Continued*

| Reagent type (species) or resource | Designation | Source or reference | Identifiers | Additional information |
|---|---|---|---|---|
| Chemical compound, drug | PicoGreen | Thermo Fisher Scientific | Cat: #P11496 | IF (1:1000) |
| Chemical compound, drug | FuGENE 6 Transfection Reagent | Promega | Cat: #E2691 | |

## Cell culture

OVCAR8, OVCAR3, MDA-MB-231, and MDA-MB-468 cells were purchased from the ATCC or the NCI Development Therapeutics Program without further authentication. Cell lines were obtained from the indicated vendors and authenticated by the suppliers prior to distribution. All experiments were performed using early-passage cells to avoid genetic drift and phenotypic variation. Cell cultures were routinely monitored for mycoplasma contamination using the MycoAlert Mycoplasma Detection Kit (Lonza) according to the manufacturer's instructions. All cell lines were maintained at 37 °C and 5% $CO_2$. Cells were cultured in RPMI1640 medium (Invitrogen) supplemented with 10% fetal bovine serum (FBS, Invitrogen). Cells were routinely tested for mycoplasma contamination using Mycoplasma Plus PCR Primer Set (Agilent, Santa Clara, CA).

## Drug combination screening and validation

OVCAR8 and MDA-MB-231 cells were used for drug combination screening. Before screening, each epigenetic inhibitor (n=74) was arrayed in 48-well plates at concentrations of 0.1 μM, 1 μM, 10 μM for IC50 calculation. For screening, cells were seeded in 48-well plates at a concentration of 3750 cells/mL for OVCAR8 and 625 cells/mL for MDA-MB-231 in 400 μL of medium per well, and then treated with a single drug or with a combination of olaparib and one of the 74 epigenetic drugs 10 days for OVCAR8 or 8 days for MDA-MB-231. When testing for synergy of olaparib with epigenetic drugs, the drug combinations were plated with five-point serially constant twofold ratio dilutions in 48-well plates using Eppendorf Xplorer plus Electronic Single Channel Pipette. Drugs were changed every two days. Cell viability testing was performed using the crystal violet staining assay (Sigma). After crystal violet was dissolved in 2% SDS, absorbance value was measured at 590 nm with spectrophotometer. Combination index (CI) and fraction affected (Fa) values were calculated using Compusyn software (*Chou, 2010*). The synergistic action between PRMTis and PARPis was further validated in a panel of cancer cell lines. In addition, a triple drug combination was also evaluated in the same cell lines. Detailed combination condition was listed in *Supplementary file 1I*.

## Colony formation assay

Cells were seeded in 48-well plates at a concentration of 3750 cells/mL for OVCAR8, 10,000 cells/mL for OVCAR3, 625 cells/mL for MDA-MB-231 and 10,000 cells/mL for MDA-MB-468 in 400 μL of medium per well overnight. Cells were then treated with a single drug or a drug combination. After fixation with methanol, colonies were stained with 0.5% crystal violet for 30 min. For quantification, crystal violet was dissolved in 2% SDS and read by microplate reader.

## CRISPR/Cas9 knockout

LentiCRISPRv2 was purchased from Addgene (plasmid #98290). sgRNAs targeting *PRMT1*, *PRMT5*, and *ERCC1* were ordered from IDT and cloned to LentiCRISPRv2. LentiCRISPRv2 and packaging vectors were transfected into 293T cells. The medium was changed 8 hr after transfection, and the medium containing lentivirus was collected 48 hr later. Cancer cells were infected with lentivirus in the presence of 8 μg/mL polybrene. 24 hr post-infection, infected cells were selected with 1 μg/mL puromycin and the surviving pools were expanded for the following experiments.

## Protein isolation and western blot

Western blotting was performed using the following primary antibodies: anti-PRMT1 (Cat No: 2449 S, CST); anti-PRMT5 (Cat No: 79998 S, CST); anti-aDMA (Cat No: 13522 S, CST); anti-sDMA (Cat No: 13222 S, CST); anti-ERCC1 (Cat No: 3885 S, CST); anti-phospho-H2AX (S139) (Cat No: 05–636, Clone

No: JBW301, Millipore); p-TBK1 (Cat No: 5483 S, CST) and anti-β-Tubulin (Cat No: 2128, Clone No: 9F3, CST), followed by incubation with secondary antibodies conjugated with horseradish peroxidase (HRP, GE Healthcare Life Sciences). Immunoreactive proteins were visualized using the LumiGLO chemiluminescent substrate (Cell Signaling).

## Comet assays

For PRMT1/5 inhibitors treatment comet assay, OVCAR8 cells were treated with DMSO, 5 µM olaparib, 5 µM GSK3368715, 5 µM olaparib combined with 5 µM GSK3368715, 5 µM GSK3235025, and 5 µM olaparib combined with 5 µM GSK3368715 for 96 hr. OVCAR3 cells were treated with DMSO, 1 µM olaparib, 1 µM GSK3368715, 1 µM olaparib combined with 1 µM GSK3368715, 1 µM GSK3235025, and 1 µM olaparib combined with 1 µM GSK3368715 for 96 hr. MDA-MB-231 cells were treated with DMSO, 5 µM olaparib, 2.5 µM GSK3368715, 5 µM olaparib combined with 2.5 µM GSK3368715, 5 µM GSK3235025, and 5 µM olaparib combined with 5 µM GSK3368715 for 96 hr. MDA-MB-468 cells were treated with DMSO, 2.5 µM olaparib, 2.5 µM GSK3368715, 2.5 µM olaparib combined with 2.5 µM GSK3368715, 5 µM GSK3235025, and 2.5 µM olaparib combined with 5 µM GSK3368715 for 96 hr. For the PRMT1/5 KO comet assay, wild type, PRMT1 sgRNA KO, and PRMT5 sgRNA KO cells were treated with 5 µM olaparib for 96 hr. Neutral comet assays with SYBR Gold staining (Invitrogen) were performed. The quantification of tail DNA was done using CASP software.

## Immunofluorescent staining

For CPD foci staining, DMSO, 2.5 µM GSK3368715 (MDA-MB-231) or 5 µM GSK3368715 (OVCAR8), 5 µM GSK3235025 treated cells or *PRMT1/5* KO cells were irradiated with UV (15 J/m$^2$) and then allowed to repair for indicated times in fresh medium. Cells were fixed in a solution containing 3% paraformaldehyde and 2% sucrose for 10 min at room temperature. Cells were subsequently permeabilized with 0.5% Triton solution for 5 min at 4 °C and then incubated with anti-CPD antibody (Cat No: CAC-NM-DND-001, Cosmo Bio LTD) in PBST buffer (PBS plus 0.1% Tween-20, 0.02% NaN3) overnight at 4 °C. Cells were then washed three times with PBST and then incubated with goat anti-mouse IgG cross-adsorbed secondary antibody, Alexa Fluor 488 (Cat No: A-11017, Thermo) for 1 hr at room temperature. After four washes with PBST, coverslips were mounted onto glass slides using Vectashield mounting medium containing DAPI (Vector Laboratories) and visualized using an Axiovert 200 M inverted microscope (Zeiss). For PicoGreen micronuclei staining, DMSO, 2.5 µM GSK3368715 (MDA-MB-231) or 5 µM GSK3368715 (OVCAR8), 5 µM GSK3235025 treated cells were fixed in solution containing 3% paraformaldehyde and 2% sucrose for 10 min at room temperature. Cells were subsequently permeabilized with 0.5% Triton solution for 5 min at 4 °C and then incubated with PicoGreen (1:1000 dilution) (Cat No: P11496, ThermoFisher Scientific) in PBST buffer (PBS plus 0.1% Tween-20, 0.02% NaN3) overnight at 4 °C. After four washes with PBST, coverslips were mounted onto glass slides using Vectashield mounting medium containing DAPI (Vector Laboratories) and visualized using an Axiovert 200 M inverted microscope (Zeiss).

## RNA isolation and qRT-PCR

Total RNA was extracted using TRIzol Reagent (Invitrogen) and reverse transcribed using the High Capacity RNA-to-cDNA Kit (Applied Biosystems). cDNA was quantified by an ABI ViiA 7 System (Applied Biosystems). QPCR primers were listed in *Supplementary file 1J*.

## RNA-sequencing

MDA-MB-231 cells were treated with DMSO, olaparib (5 µM, 6 days), GSK3368715 (2.5 µM, 6 days), GSK3235025 (5 µM, 6 days) or combinations. Following total RNA extraction, libraries were prepared using TruSeq Stranded mRNA Library Prep Kit (Illumina), and equimolar libraries were multiplexed and sequenced on an Illumina NextSeq 500 (pair end 100 bp reads) by the BGI Genomics. FASTQ files were downloaded for the analysis.

## Splicing analysis

Paired-end RNA-seq profiles were used to characterize treatment-induced splicing changes. Differential splicing was detected by the rMATS algorithm, using a hierarchical framework to model exon inclusion levels (*Shen et al., 2014*; *Wang et al., 2017*). Five alternative splicing events were identified,

including skipped exon (SE), alternative 5' splice site (A5SS), alternative 3' splice site (A3SS), mutually exclusive exons (MXE), and retained intron (RI).

## ERCC1 overexpression

For ERCC1 overexpression, OVCAR8 and MDA-MB-231 cells were seeded in 12-well plates at a confluence of 50% one day before transduction. Cells were transduced with lentiviral CD513B vector (System Biosciences) or CD513B-ERCC1. Medium was changed the next day. Cells were selected with 5 µg/mL puromycin for 3 days, then seeded for sensitivity detection of olaparib alone, GSK3368715 alone /GSK3235025 alone, or combination of olaparib with GSK3368715 or GSK3235025.

## DNA dot-blot

Cells were irradiated with UV (15 J/m$^2$) and then allowed to repair for indicated times in fresh medium. Genomic DNA was extracted from these UV-irradiated cells using the QIAamp DNA Blood Mini Kit (51104, Qiagen) and RNase A (19101, Qiagen). DNA solutions were prepared in 50 µL of PBS per sample to have 100 ng, 50 ng, and 25 ng of genomic DNA. DNA solutions were spotted onto Hybond-N+positively charged nylon membrane (RPN1210B, GE Healthcare). Membranes were blocked at room temperature for 1 hr in TBS-T (Tris-Buffered Saline (TBS), 0.05% Tween-20) containing 5% skim milk and incubated with the anti-CPD antibody. Membranes were washed four times in TBS-T, then incubated for 1 hour with the appropriate peroxidase-conjugated anti-mouse secondary antibody at 1/5,000 (EMD Millipore). Immunoreactive proteins were visualized using the LumiGLO chemiluminescent substrate (Cell Signaling).

## Luciferase assays

-1500 ~ +120 bp of *ERCC1* promoter was cloned into the pGL3-basic reporter vector (Promega) to make an *ERCC1*-P-Luc luciferase reporter plasmid. For drug treatment reporter assay, 500 ng of reporter vector plus 50 ng of the renilla luciferase plasmid was transfected to cells using FuGENE 6 (Promega); cells were then treated with DMSO, 5 µM GSK3368715, and 5 µM GSK3235025 for 48 hr. For the *PRMT1* and *PRMT5* CRISPR KO reporter assay, cells infected with control, *PRMT1*-sgRNA, and *PRMT5*-sgRNA lentivirus were then transfected with reporter vector and renilla, and cells were harvested 48 hr after transfection. Reporter assays were performed using a dual luciferase reporter assay system (Promega) by Fluoroskan Ascent FL fluorometer (Thermo Fisher Scientific).

## In vivo treatment experiment

For MDA-MB-231 xenograft experiments, three million tumor cells were injected subcutaneously into 6-week-old athymic female nu/nu mice (Stock No: 002019, Jackson Labs) and grown until tumors reached a size of approximately 30 mm$^3$. Xenografted mice were randomized and then received vehicle, 50 mg/kg olaparib ip, 25 mg/kg GSK3368715 po, 50 mg/kg GSK3235025 po or the combination of two agents and three agents, 5 days a week for 3 weeks (n=6–8 per group). Caliper measurements were taken every three days starting from the initiation of drug treatment. Tumor volumes were calculated according to the formula: tumor volume [mm$^3$] = (1/6) × $\pi$ × (tumor length) × (tumor width)$^2$. Due to the nature of the performed experiments, no randomization and no blinding was used because it was deemed unfeasible. However, the resulting tumors were analyzed in a blinded manner. All animal procedures were in accordance with protocols (#807557) approved by the Institutional Animal Care and Use Committee of the University of Pennsylvania.

## Estimation of impact score of siRNA experiment

For a specific gene *g*, we used $exp(t)_g$ to represent its expression measured by RNA-seq in a cell line treated with a given drug *t*, and $expC_g$ to represent its expression in a control treatment. Fold changes of expression were calculated for a specific gene by comparing its expression in each treatment condition to control treatment,

$$Fexp(t)_g = \log_2\left(\exp(t)_g\right) - \log_2\left(\exp C_g\right).$$

For a specific gene *g*, we used $IC50(k)_g$ to represent IC50 of PARPi treatment in a cell line after knockdown of this gene by pooled siRNA transfection, and $IC50C$ to represent IC50 of PARPi

treatment in a control siRNA transfection. Fold changes of IC50 were calculated for a specific gene knockdown by comparing PARPi IC50 in the same cell line in siRNA knockdown condition to control siRNA transfection,

$$FIC50\,(k)_g = \frac{\text{IC50}\,(k)_g}{\text{IC50C}}.$$

Fold changes of IC50 were then normalized by min-max feature scaling,

$$F'IC50\,(k)_g = \frac{FIC50\,(k)_g - FIC50\,(k)_{min}}{FIC50\,(k)_{max} - FIC50\,(k)_{min}}.$$

A final priority score was calculated for each gene examined,

$$Score_g = F'IC50\,(k)_g * \sum Fexp\,(t)_g.$$

### TCGA RNA-seq data processing and gene expression analysis

The poly(A)+RNA seq data for primary tumors and their adjacent tissues were generated by the University of North Carolina and the British Columbia Cancer Agency Genome Sciences Centre as part of the TCGA project. All RNA-seq data were processed through a pipeline developed by the UCSC Toil RNAseq Recompute Compendium, which allowed us to consistently process large-scale RNA-seq data without computational batch effects (*Vivian et al., 2017*). For TCGA RNA-seq data, if more than one sample existed for a participant, one single tumor sample (and matched adjacent sample, if applicable) was selected based on the following rules: (1) tumor sample type: primary (01)>recurrent (02)>metastatic (06); (2) order of sample portions: higher portion numbers were selected; and (3) order of plate: higher plate numbers were selected.

### TCGA mRNA differential expression meta-analysis

For each cancer type which had at least three normal tumor-adjacent tissues in TCGA, differential gene expression analysis between tumor tissues and normal tumor-adjacent tissues was performed using one-sided *t*-test (for up-regulation and down-regulation respectively). p Values across all cancer types were then combined via Fisher's combined probability test (*Mosteller and Fisher, 1948*) to yield a meta-p value to assess the degree of up-regulation or down-regulation at the pan-cancer level. The meta-p value on the direction with the more significant dysregulation was reported for the corresponding PRMT genes.

### CPTAC proteomic data processing and protein expression analysis

Extensive mass spectrometry-based proteomics data using isobaric tagging approaches (iTRAQ or TMT) for selected cancer types were generated by the National Cancer Institute's CPTAC. Protein-level processed data consisting of iTRAQ or TMT log(ratios) were downloaded from the CPTAC data portal (https://cptac-data-portal.georgetown.edu) on January 28, 2021.

### CPTAC protein differential expression meta-analysis

Differential expression analysis between tumor tissues and paired normal tissues was performed using one-sided *t*-test (for up-regulation and down-regulation respectively). p Values across all cancer types were then combined via Fisher's combined probability test (*Mosteller and Fisher, 1948*) to yield a meta-p value to assess the degree of up-regulation or down-regulation at the pan-cancer level. The meta-p value on the direction with the more significant dysregulation was reported for the corresponding PRMT.

### TCGA survival meta-analysis

For each PRMT gene in a given cancer type, we fit a Cox proportional hazards model and obtained a z-statistic and a p-value (with direction of effect) to assess whether high expression was associated with favorable or unfavorable overall survival. p-values across all cancer types were then combined via Fisher's combined probability test (*Mosteller and Fisher, 1948*) to yield a meta-p value to assess the prognostic value at pan-cancer levels (for risky effect and protective effect separately). The meta-p

value on the direction with the more significant prognostic association was reported for the corresponding PRMT gene.

## Recurrent genomic analysis

Recurrent somatic copy number alterations and mutations of the PRMT family were analyzed in the TCGA sample cohort by a standard pipeline developed by the Functional Cancer Genome project (*Jiang et al., 2022*; *Hu et al., 2021*; *Hu et al., 2019*). The G-score (somatic copy number alterations) and M-score (mutations) were established for each PRMT gene at both individual and pan-cancer levels.

## Statistical analysis

Large-scale and multi-dimensional profiling data generated by the publicly accessible databases (TCGA, CPTAC, and DepMap) were used; therefore, statistical analysis was not used to predetermine sample size in this study. For TCGA analysis, if more than one profiling file existed for a patient in TCGA, only one single file would be selected and used, and detailed methods for exclusion of duplicated profiling files are described in the method section. The computational analyses were not randomized, and the investigators were not blinded during data analyses of this study. When applicable, enrichment was tested using Fisher's exact test with FDR correction. Cell viability and gene relative expression data were shown as means with standard deviation (SD). Comparisons between groups were performed using the student t-test.

## Acknowledgements

Research reported in this publication was supported by the National Cancer Institute of the National Institutes of Health (R01CA285598). The content is solely the responsibility of the authors and does not necessarily represent the official views of the National Institutes of Health. LZ was also supported by the Basser Center for BRCA and the National Institutes of Health grants (R01CA262070 and R01CA288850). ZH and LZ were supported by the Adenoid Cystic Carcinoma Research Foundation and Torrey Coast Foundation. XH was supported by the Ovarian Cancer Research Alliance. XH and YZ were supported by the Foundation for Women's Cancer. Support of the core facilities was provided by a Cancer Center Support Grant (P30CA016520) to Abramson Cancer Center.

## Additional information

### Competing interests

Youyou Zhang: Y.Z. is an employee of GlaxoSmithKline. Omid Tavana: O.T. is an employee of AstraZeneca. Robert H Vonderheide: R.H.V. is an inventor on licensed or pending patents relating to cancer cellular immunotherapy and receives royalties from Children's Hospital Boston for a licensed research-only monoclonal antibody. Ho Man Chan: H.M.C. is an employee of AstraZeneca. Lin Zhang: L.Z. reports having received research funding from AstraZeneca, Bristol-Myers Squibb/Celgene, and Prelude Therapeutics. Xiaowen Hu: X.H. reports having received research funding from AstraZeneca, Bristol-Myers Squibb/Celgene, and Prelude Therapeutics. The other authors declare that no competing interests exist.

### Funding

| Funder | Grant reference number | Author |
| --- | --- | --- |
| National Cancer Institute | R01CA262070 | Lin Zhang |
| National Cancer Institute | R01CA285598 | Lin Zhang |
| National Cancer Institute | R01CA288850 | Lin Zhang |
| Ovarian Cancer Research Alliance | | Xiaowen Hu |

| Funder | Grant reference number | Author |
|---|---|---|
| Foundation for Women's Cancer | | Youyou Zhang<br>Xiaowen Hu |
| Adenoid Cystic Carcinoma Research Foundation | | Zhongyi Hu |
| Torrey Coast Foundation | | Lin Zhang |

The funders had no role in study design, data collection and interpretation, or the decision to submit the work for publication.

## Author contributions

Youyou Zhang, Mu Xu, Conceptualization, Data curation, Formal analysis, Investigation, Visualization, Writing – original draft, Writing – review and editing; Jiao Yuan, Investigation, Visualization, Methodology; Zhongyi Hu, Data curation, Investigation, Visualization, Methodology; Junjie Jiang, Yanrong Sun, Jie Huang, Bingwei Wang, Investigation; Yuxin Wang, Visualization; Jianfeng Shen, Resources; Meixiao Long, Yi Fan, Resources, Data curation; Kathleen T Montone, Design of epigenetic inhibitor combinations in preclinical models and provided expertise in clinical oncology; Janos Tanyi, Design of epigenetic inhibitor combinations in preclinical models and provided expertise in clinical oncology; Sarah H Kim, Design of epigenetic inhibitor combinations in preclinical models and provided expertise in clinical oncology; Omid Tavana, Design of epigenetic inhibitor combinations in preclinical models and provided expertise in early-stage drug development; Robert H Vonderheide, Design of epigenetic inhibitor combinations in preclinical models and provided expertise in clinical oncology; Ho Man Chan, Design of epigenetic inhibitor combinations in preclinical models and provided expertise in early-stage drug development; Susan Domchek, Design of epigenetic inhibitor combinations in preclinical models and provided expertise in clinical oncology; Lin Zhang, Conceptualization, Writing – original draft, Writing – review and editing; Xiaowen Hu, Conceptualization, Formal analysis, Visualization, Writing – original draft, Writing – review and editing

## Author ORCIDs

Lin Zhang ⓘ https://orcid.org/0000-0003-1998-0611
Xiaowen Hu ⓘ https://orcid.org/0000-0002-3269-4356

## Ethics

All animal procedures were in accordance with protocols (#807557) approved by the Institutional Animal Care and Use Committee of the University of Pennsylvania.

Reviewer #2 (Public review): https://doi.org/10.7554/eLife.99225.4.sa1
Author response https://doi.org/10.7554/eLife.99225.4.sa2

# Additional files

## Supplementary files

Supplementary file 1. Supplementary tables A to J. (**A**) List of the epigenetic modulators that were used in initial drug combination screen. (**B**) List of the specimens that were used in this study from the TCGA project. (**C**) Expression of the PRMTs across cancers. (**D**) Recurrent genomic alterations of the PRMTs across cancers. (**E**) Cell dependencies of PRMTs from the DepMap project. (**F**) Correlations between the expression levels of *PRMT1/5* and "50 hallmark" molecular signatures across the TCGA tumor specimens. (**G**) List of the potential BRCAness genes in the human genome. (**H**) List of the BRCAness genes that were repressed by PRMTi treatment. (**I**) PRMTi and PARPi combination concentration in each cancer line. (**J**) Primers and oligoes that were used in this study.

MDAR checklist

## Data availability

The genomic profiles of human cancers were generated by the TCGA project, which are publicly available through the Genomic Data Commons portal (GDC, https://portal.gdc.cancer.gov/). Protein-level processed data consisting of iTRAQ or TMT log(ratios) were downloaded from the CPTAC data

portal (https://cptac-data-portal.georgetown.edu). Genetic screening profiles in human cancer cell lines were generated by the DepMap and the Score projects, which are publicly available through the DepMap portal (https://depmap.org/portal/), and the Score projects (https://score.depmap.sanger.ac.uk/). The genomic data were retrieved, processed and analyzed through a master computational protocol developed by the Functional Cancer Genome project as described by our previous publications (*Jiang et al., 2022*; *Hu et al., 2021*; *Hu et al., 2019*) as well as the method section. Raw sequencing data have been deposited in NCBI BioProject under accession PRJNA945030. This paper does not report original code.

The following dataset was generated:

| Author(s) | Year | Dataset title | Dataset URL | Database and Identifier |
|---|---|---|---|---|
| Zhang Y, Hu Z, Zhang L, Hu X | 2023 | Repression of PRMT activities sensitize homologous recombination-proficient ovarian and breast cancer cells to PARP inhibitor treatment | https://www.ncbi.nlm.nih.gov/bioproject/PRJNA945030 | NCBI BioProject, PRJNA945030 |

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
