## [Editor Report · eLife Assessment]

This study presents a **valuable** and interesting finding that a combination of arginine methyltransferase inhibitors synergize with PARP inhibitors to eliminate ovarian and triple negative breast cancer cell lines in vitro and in vivo using preclinical mouse models. The data were collected and analyzed using **solid** and validated methodology and can be used as a starting point for the development of novel therapeutics. The work will be of broad interest to scientists working in the field of breast cancer and ovarian cancer.

---

## [Referee Report · Reviewer #2 (Public review)]

Summary:

The authors show that a combination of arginine methyltransferase inhibitors synergize with PARP inhibitors to kill ovarian and triple negative cancer cell lines in vitro and in vivo using preclinical mouse models.

Strengths and weaknesses

The experiments are well-performed, convincing and have the appropriate controls (using inhibitors and genetic deletions) and use statistics.

They identify the DNA damage protein ERCC1 to be reduced in expression with PRMT inhibitors. As ERCC1 is known to be synthetic lethal with PARPi, this provides a mechanism for the synergy. They use cell lines only for their study in 2D as well as xenograph models.

Comments on revisions:

The authors have addressed by final concerns.

---

## [Author Response]

The following is the authors’ response to the previous reviews

**Reviewer #2 (Public Review):**
Summary:The authors show that a combination of arginine methyltransferase inhibitors synergize with PARP inhibitors to kill ovarian and triple negative cancer cell lines in vitro and in vivo using preclinical mouse models.Strengths and weaknessesThe experiments are well-performed, convincing and have the appropriate controls (using inhibitors and genetic deletions) and use statistics.They identify the DNA damage protein ERCC1 to be reduced in expression with PRMT inhibitors. As ERCC1 is known to be synthetic lethal with PARPi, this provides a mechanism for the synergy. They use cell lines only for their study in 2D as well as xenograph models.

We sincerely thank Reviewer #2 for the insightful and constructive feedback, as well as for the kind recognition of the scientific quality of our work: “The experiments are well-performed, convincing and have the appropriate controls (using inhibitors and genetic deletions) and use statistics.” We sincerely thank Reviewer #2 for their thoughtful and constructive comments during both rounds of review, which have significantly improved the quality of our manuscript. In response, we have incorporated new results from additional experiments into the figures (Figures 6M and 6N) and made comprehensive revisions throughout the text, figures, and supplementary materials. Following the reviewer’s valuable suggestions, we also revised the Discussion section. In the “Recommendations for the authors” sections, we have provided detailed point-by-point responses to each comment, which were instrumental in guiding our revisions. We believe these updates have substantially strengthened the manuscript and fully addressed all reviewer concerns.

**Reviewer #2 (Recommendations for the authors):**
Although the authors have addressed each recommendation from the reviewer, further revision of the manuscript are still necessary, as outlined below.Add these additional comments in the text to further enhance the comprehension and clarity of the data.(1) If the authors kept the tumors of various sizes in Figure 7I, it would be important to assess the protein and/or mRNA level of ERCC1 to further support their mechanism.Question (1): Please add the figures of new experiments (treatment diagram, curves for tumor volume and qRT-PCR data) to Figure 6.

We thank the reviewers for their constructive suggestions. In response to the reviewers’ comments, we have added the treatment diagram and qPCR results to Figure 6. In this experiment, we shortened the treatment duration to seven days to assess early molecular responses to therapy rather than downstream effects. As expected, such short-term treatment did not result in significant differences in tumor growth among groups. The new results are now presented in Figure 6, panels M and N. The corresponding results and figure legends will also be included in the revised version of the manuscript

(2) Figure 2G: please explain why two bands remain for sgPRMT1.Question (2): In the answer, the authors stated, "Upon knockdown of the major isoforms by CRISPR/Cas9, expression of this minor isoform may have increased as part of a compensatory feedback mechanism, rendering it detectable by immunoblotting." Please put the statement into the discussion section.

We sincerely thank the reviewers for their thoughtful and constructive suggestions. In response to these comments, we have carefully revised the manuscript and incorporated the corresponding information into the Discussion section to provide greater clarity and context for our findings.

(3) (Previously point 5) What is the link with ERCC1 splicing because reduced overall ERCC1 expression is clear?Question (5): Please add the explanation you provide of links between ERCC1 splicing and PRMTi into the discussion section."Furthermore, as shown in Figure 4G, we observed a reduction in the total ERCC1 mRNA reads following PRMTi treatment. This decrease may be attributed, at least in part, to the instability of the alternatively spliced ERCC1 transcripts, which could be more prone to degradation. In combination with the transcriptional downregulation of ERCC1 induced by PRMT inhibition, these alternative splicing events may lead to a further reduction in functional ERCC1 protein levels. This dual impact on ERCC1 expression, through both decreased transcription and the generation of unstable or nonfunctional isoforms, likely contributes to the enhanced cellular sensitivity to PARP inhibitors observed in our study."

We sincerely thank the reviewers for their thoughtful and constructive suggestions. In response to these comments, we have carefully revised the manuscript and incorporated the corresponding information into the Discussion section to provide greater clarity and context for our findings.

(4) (Previously 6) Figure 7J: From the graph, it seems like Olaparib+G715 and G715+G025 have a similar effect on tumor volume (two curves overlap). Please discuss.Question (6): In the answer, the authors stated, "Our in vitro and in vivo findings, together with previously published data, consistently demonstrate that GSK715 is more potent than both GSK025 and Olaparib. Notably, treatment with GSK715 alone led to significantly greater inhibition of tumor growth compared to either GSK025 or Olaparib administered individually. This higher potency of GSK715 also explains the comparable levels of tumor suppression observed in the combination groups, including GSK715 plus Olaparib and GSK715 plus GSK025. These results suggest that GSK715 is likely the primary driver of efficacy in the two drug combination settings." Please put the statement in the corresponding result section for Figure 6J.

We sincerely thank the reviewers for their thoughtful and constructive suggestions. In response to these comments, we have carefully revised the manuscript and incorporated the corresponding information into the result section for Figure 6J to provide greater clarity and context for our findings.